# 🔥 FiRE: Fine-grained Ranking Evaluation for Machine Translation

## Abstract

Developing reliable machine translation (MT) systems hinges on our ability to distinguish superior translations from inferior ones. However, existing evaluation paradigms, whether limited to coarse overall rankings or misaligned with human preferences, fail to deliver interpretable, fine-grained feedback in reference-free settings. We present a **Fi**ne-Grained **R**anking **E**valuation method (**FiRE**) that leverages off-the-shelf large language models to perform criterion-driven pairwise comparison across three complementary dimensions: faithfulness, fluency, and consistency of style, instead of producing a single holistic judgment. To enable rigorous meta-evaluation of evaluation paradigms in the absence of any suitable testbed, we construct the first human-annotated, reference-free benchmark for fine-grained ranking evaluation, achieving substantial inter-annotator agreement. Through meta-evaluation on this benchmark, FiRE demonstrably outperforms leading regression-based and error-analysis metrics in aligning with human comparative judgments, while providing more informative insights into translation quality. Finally, our examination of LLM evaluator biases (position and self-enhancement) and their handling of tied cases offers guidance for more nuanced MT evaluation.

## 1 Introduction

The goal of machine translation (MT) is to produce high-quality translations that align with human preferences, so progress therefore hinges on reliably distinguishing better outputs from worse ones. Large language models (LLMs) exhibit strong multilingual and generation capabilities, and their translations often satisfy the classical desiderata of accuracy and fluency. In this high-quality regime, traditional overlap-based metric BLEU (Papineni et al., 2002) and regression-based metrics such as BERTScore (Zhang et al., 2019) are frequently insufficiently discriminative (Freitag et al., 2022). Ranking-based evaluation, introduced in other generation tasks (Wang et al., 2023; Chiang et al., 2024) and recently adapted to MT (Ibraheem et al., 2024), improves separability over regression-based metrics but still lacks interpretability.

Consider the illustrative case in Figure 1: two translations (T1, T2) are comparable in overall quality. T1 is more formal and closer to the source's style, while T2 reads more fluently. A vanilla ranking method that outputs only a better or worse decision may label one as superior, yet it provides no rationale for the trade-off across criteria. In contrast, error-based evaluation provides rich diagnostic information by identifying and categorizing errors. Motivated by this, we introduce **FiRE**, a novel **Fi**ne-grained **R**anking **E**valuation framework. FiRE performs criterion-based, reference-free pairwise comparison by making explicit judgments on faithfulness, fluency, and consistency of style, and then synthesizes them into an overall decision. The pairwise setting enhances sensitivity to subtle differences, and the explicit criteria make outcomes more transparent and explainable.

A significant hurdle in developing and validating such fine-grained evaluation methods has been the lack of suitable benchmarks. While existing benchmarks, as detailed in Section 2.2, have utilized relative rankings, inferred preferences from scalar scores, or employed contrastive perturbations, a critical gap remains for a benchmark offering direct human pairwise ranking across multiple explicit criteria in a reference-free setting for modern MT. To overcome this, we introduce the first human-annotated benchmark specifically designed for reference-free, fine-grained pairwise ranking evaluation. Our benchmark comprises two translation directions, English-to-Chinese and Russian-

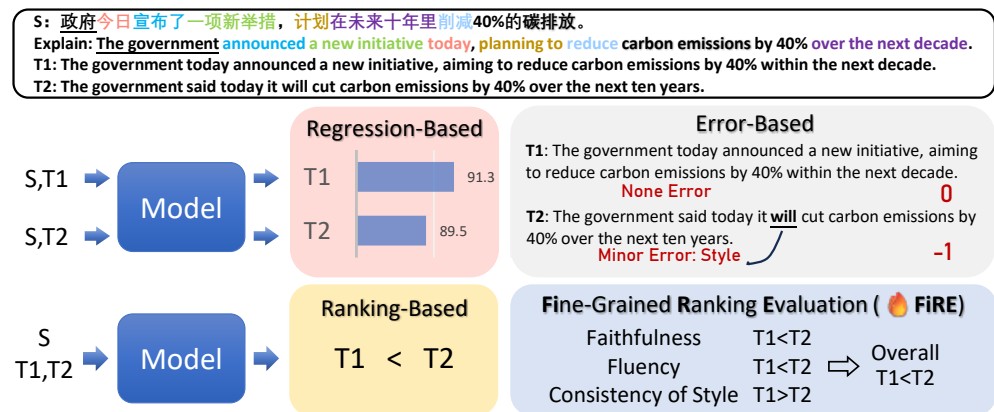

Figure 1: Illustrative case of regression-based, error-based, ranking-based evaluation, and our proposed fine-grained ranking evaluation (FiRE).

to-Chinese, each containing $1,600$ data points. Each data point consists of one source sentence, two translation candidates, and four human annotations across four criteria (faithfulness, fluency, consistency of style, and overall quality), without access to reference translations, yielding a total of 12,800 annotations.

We conduct extensive experiments using FiRE with seven LLMs—four open-source (DeepSeek-R1, QwQ-32B, Mistral-Large-Instruct, Qwen2.5-72B-Instruct) and three closed-source (GPT-4o, Claude-3.5-Sonnet, Gemini-2.0-Flash)—and compare it against established evaluation paradigms. FiRE delivers clear advantages over regression- and error-based evaluation by enabling side-by-side comparison of two translations, which simplifies evaluation, surfaces nuanced differences, and improves decision accuracy. Synthesizing the fine-grained judgments into a single overall decision yields additional gains, indicating that criterion-aware aggregation leverages complementary evidence across faithfulness, fluency, and consistency of style, reduces noise on near-tie cases, and better reflects how humans trade off these factors in overall judgments. At the criterion level, FiRE surpasses error-based evaluation, achieving higher agreement on faithfulness, fluency, and consistency of style, which in turn yields stronger overall rankings.

We use DeepSeek-R1 as FiRE backbone to evaluate six MT systems and compare the results with other metrics. Consistent with other reference-free evaluations, FiRE identifies GPT-4o as the top-performing model, further revealing its superior performance across nearly all evaluation criteria. In a notable departure, while ALMA-13B-R (Xu et al.) achieves a top ranking on metrics like COMET-Kiwi (Rei et al., 2023), FiRE exposes a critical weakness: the faithfulness score of ALMA-13B-R is the second-lowest among all tested models, which indicates a strong tendency for hallucination. Furthermore, unlike holistic scores that can mask performance variations, FiRE can discern model strengths across different translation directions, identifying that DeepL excels in English-to-Chinese while LanMT performs better in Russian-to-Chinese. We show that FiRE supports actionable system-level diagnosis by revealing where models gain or lose across faithfulness, fluency, and stylistic consistency, clarifying directional strengths that holistic scores obscure. To our knowledge, this is the first work to use LLMs for fine-grained and overall pairwise ranking evaluation of machine translation.

## 2 RELATED WORK

### 2.1 PARADIGMS OF MT EVALUATION

MT evaluation methodologies can be broadly classified into regression-based, error-based, and ranking-based approaches.

**Regression-Based Evaluation.** This paradigm assesses translation quality by assigning a scalar score. BLEU (Papineni et al., 2002) is the dominant overlap-based metric that measures quality by computing n-gram precision between machine-generated translations and reference translations.

Due to its lack of ability to capture semantic features, researchers introduce regression-based metrics, including BERTScore (Zhang et al., 2019), COMET (Rei et al., 2020), BLEURT (Sellam et al., 2020), and MetricX-24-XXL (Juraska et al., 2024), which use pre-trained language models to predict scores for the quality of translation, focusing on semantic similarity. This paradigm offers simple scalar scores but remains coarse-grained.

**Error-Based Evaluation.** The Multidimensional Quality Metrics (MQM) framework (Lommel, 2013) is a widely used method for assessing translation quality by identifying and categorizing various errors. MQM-based metrics enable a more nuanced and interpretable evaluation of MT systems, aligning closely with human judgment by pinpointing specific translation errors and their severity. Recent advancements have sought to automate or semi-automate this process using pre-trained language models (PLMs), leading to metrics like xCOMET (Guerreiro et al., 2023), which incorporates error span detection, and LLM-driven systems such as GEMBA-ESA (Kocmi & Federmann, 2023b), GEMBA-MQM (Kocmi & Federmann, 2023a), EAPrompt (Lu et al., 2023), and M-MAD (Feng et al., 2024). Though their primary focus is error diagnosis, they can also produce a numeric score by aggregating error types and numbers. This paradigm provides interpretable error diagnostics, but its aggregated scores are not tailored for direct pairwise ranking.

**Ranking-Based Evaluation.** This paradigm directly compares two or more translation candidates for a given source segment and determines their relative order of quality. Ye et al. (2007) formulate MT evaluation as this ranking problem, typically in a reference-based setting. Subsequent research explores various features and learning algorithms to improve ranking accuracy, sometimes aiming to reduce reliance on full reference translation (Duh, 2008; Guzmán et al., 2014; 2015; Song & Cohn, 2011; Zhang & van Genabith, 2020). Recognizing that human-labeled references are scarce or unavailable in practical scenarios, reference-free ranking methods have gained traction. For instance, MT-Ranker (Ibraheem et al., 2024) employs a multi-stage training regime to develop a specialized model for reference-free ranking. This paradigm aligns naturally with choosing the better translation but existing methods remains coarse-grained, producing only a holistic overall ranking.

## 2.2 Pairwise Ranking Evaluation Benchmarks in MT

Since MT evaluation was first formalized as ranking problem (Ye et al., 2007), a variety of datasets have been employed to conduct meta-evaluations of pairwise ranking methods. Early research relied on the relative ranking datasets from WMT shared tasks between 2008 and 2016 (RR08–16), in which five translation candidates for each source sentence are ranked from best to worst by human annotators with reference to a gold-standard translation (Callison-Burch et al., 2008). However, these datasets have become temporally outdated and are less reflective of modern MT systems.

Recent studies relied on synthetic pairwise datasets, which fall into two main categories: those derived from human-assigned scores and those generated through contrastive perturbations. The first category includes Direct Assessment (DA17–22) (Mathur et al., 2020) and Multidimensional Quality Metrics (MQM20–23) (Pal et al., 2023) datasets. DA datasets contain translations from multiple MT systems, each annotated with a quality score ranging from 0 to 100. In contrast, MQM datasets employ expert annotators to identify error spans with fine-grained error types and severity levels, yielding weighted error scores that reflect translation quality. The second category is exemplified by the ACES dataset (Amrhein et al., 2022), a contrastive synthetic benchmark constructed through adversarial perturbations. For each predefined error type, annotators or automated scripts introduce targeted errors into otherwise correct translations, resulting in contrastive translation pairs. These pairs are designed to test the sensitivity and robustness of evaluation metrics to specific types of translation errors. Despite these advances, a benchmark for direct human pairwise ranking on multiple explicit criteria (e.g., faithfulness, fluency, style) in a reference-free setting, especially for modern MT outputs, has been lacking.

## 3 Fine-Grained Ranking Evaluation

### 3.1 Problem Definition

In real-world scenarios, users may have diverse and multidimensional requirements for comparing translations. To simulate these preferences and provide fine-grained information accordingly, we investigate existing evaluation frameworks (Callison-Burch et al., 2007; Mirkin & Meunier, 2015;

Sun et al., 2023; Lommel, 2013) and select three widely used criteria: *faithfulness*, *fluency*, and *consistency of style*. **Faithfulness** refers to the accuracy of the translation in conveying the original meaning of the source sentence, reflecting the extent of hallucination by the translation model. **Fluency** refers to the naturalness and readability of the translation, indicating the model's ability to generate coherent and grammatically correct text in the target language. **Consistency of style** refers to the uniformity and coherence of the translation in terms of tone and style, reflecting the model's ability to maintain a consistent stylistic output across different languages. These criteria are crucial for evaluating translation quality across various dimensions and meeting diverse user needs (Kirchhoff et al., 2012; Lommel, 2013; Sun et al., 2024a). In addition, we also incorporate an **overall** quality criterion to provide a holistic view and align with the traditional ranking problem.

Given a source sentence $x$ and two translation candidates $y_1$ and $y_2$, the goal of fine-grained ranking evaluation is to determine which translation is superior according to the specific criterion $c$, which is provided by the user based on their practical needs. We denote the fine-grained ranking evaluation as $\mathcal{M}(c, x, y_1, y_2) \rightarrow p$, where $\mathcal{M}$ is the evaluator and $p \in \{y_1 \succ y_2(A), y_2 \succ y_1(B), y_1 \sim y_2(E)\}$ is the evaluation outcome, indicating whether translation $y_1$ is superior to $y_2$, $y_2$ is superior to $y_1$, or both translations are considered equally preferred according to the criterion $c$.

## 3.2 DATA COLLECTION

To construct the fine-grained ranking benchmark, we adopt six MT systems, including three open-source systems (NLLB-200-1.3B (NLLB Team et al., 2024), ALMA-13B-R (Xu et al.), Qwen2-72B-Instruct (Yang et al., 2024a)) and three closed-source systems (GPT-4o, DeepL, LanMT), to generate translation candidates for each source sentence. Details of MT systems are displayed in Appendix I. Our study focuses on two high-resource language directions: English-to-Chinese (EN→ZH) and Russian-to-Chinese (RU→ZH). We collect source sentences from the WMT23 test set (Blain et al., 2023) for the English-to-Chinese and Russian-to-English translation tasks, respectively. For each source

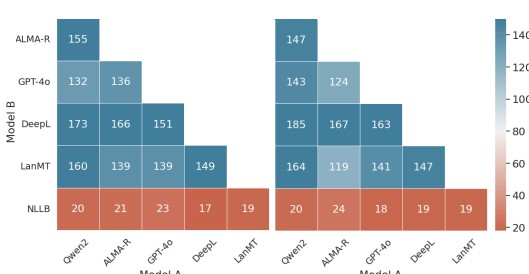

Figure 2: Number of pairwise comparison data points for each combination of MT systems, shown for EN→ZH (left) and RU→ZH (right).

sentence, we generate Chinese translation candidates using the six MT systems and construct 15 pairwise data points by enumerating all possible compositions of translation candidates. We filter out data points with identical translation candidates and sample uniformly across the pairwise data to ensure broader coverage of source sentences and a more balanced distribution of translation compositions. Since the performance of NLLB-200-1.3B is lower than that of the other MT systems, we downsample its compositions to balance annotation quality and the spectrum of benchmark (explained in Appendix J). The statistics of the fine-grained ranking data are displayed in Figure 2.

Before full-scale annotation, we conducted several pilot rounds to calibrate annotators and refine the guidelines. For each language direction, three annotators evaluate the pairwise comparisons, selecting the superior translation according to the specified criterion. Ties are allowed. Table 1 presents the annotation statistics. While a 2-annotator 2-class annotation achieves substantial agreement with $\kappa > 0.61$ (Landis & Koch, 1977), our 3-annotator 3-class setting (which typically yields lower $\kappa$ values) shows comparably substantial inter-annotator reliability ($\kappa = 0.57 - 0.81$). We believe this strong agreement arises because pairwise ranking with explicit criteria is cognitively simpler and less ambiguous than assigning absolute scores. A small number of items where three annotators select different labels for a given criterion are discarded after computing $\kappa$, since such cases do not yield a reliable consensus. As a result, the number of retained pairs differs slightly across criteria. Language proficiency of annotators is detailed in Appendix E.

We employ majority voting on the human-annotated pairwise data to assign final labels. The resulting data is categorized into two groups: ranked data, which includes cases where $y_1 \succ y_2$ or $y_2 \succ y_1$, and tied data, where $y_1 \sim y_2$. Statistics are detailed in Table 18 in the Appendix. This classification into ranked and tied sets is instrumental for conducting a more nuanced meta-evaluation of different

Table 1: Statistics of human annotations for each criterion. $\kappa$ denotes the value of Fleiss' kappa.

| | EN→ZH | | RU→ZH | |
|---|---|---|---|---|
| | # of pairs | $\kappa$ | # of pairs | $\kappa$ |
| Faithfulness | 1574 | 0.66 | 1591 | 0.81 |
| Fluency | 1574 | 0.66 | 1592 | 0.76 |
| Consistency of Style | 1568 | 0.57 | 1594 | 0.62 |
| Overall | 1574 | 0.66 | 1588 | 0.75 |

methods across the various criteria. Notably, the overall ranking exhibits the highest ranked rate. This suggests that annotators are more inclined to make a definitive better-worse judgment when assessing overall quality, potentially because this holistic evaluation integriates various aspects of translation, facilitating clearer distinctions.

## 4 EXPERIMENTS

We investigate the efficacy of several state-of-the-art LLMs as evaluators in criterion-based pairwise MT evaluation—covering faithfulness, fluency, and stylistic consistency—and overall translation quality in the EN→ZH and RU→ZH directions. We benchmark representative baseline methods from different evaluation paradigms and analyze their respective strengths and limitations.

### 4.1 BASELINES

**Regression-Based Evaluators.** We employ four reference-free evaluation models that produce a quality score, including two versions of xCOMET (Guerreiro et al., 2023) and two versions of COMET-Kiwi (Rei et al., 2023). We compare the score of each translation candidate and obtain the overall pairwise judgment.

**Error-Based Evaluators.** We adopt two LLM-as-judge approaches—M-MAD (Feng et al., 2024) and GEMBA-MQM (Kocmi & Federmann, 2023a)—in their original implementations. Error types in each method are mapped to our three evaluation criteria (see Table 13 in the Appendix for details). For each translation candidate, we aggregate the number and severity of errors under each criterion to compute an error-based score, which yields a pairwise better-worse judgment. To derive the overall pairwise judgment, we further aggregate these three scores by simple summation to obtain a single composite error-based score.

**Ranking-Based Evaluators.** We incorporate two versions of the state-of-the-art reference-free ranking-based evaluator MT-Ranker (Ibraheem et al., 2024) that function as a binary classifier.

### 4.2 LLM EVALUATORS

We use several state-of-the-art LLMs as FiRE evaluators. The set of LLM evaluators includes four open-source models: Deepseek-R1 (DeepSeek-AI, 2025), QwQ-32B (Team, 2025), Mistral-Large-Instruct, and Qwen2.5-72B-Instruct (Yang et al., 2024b), as well as three closed-source models: GPT-4o, Claude-3.5-Sonnet, and Gemini-2.0-Flash. The LLM evaluators are prompted with the source sentence and the two translation candidates, along with the specified criterion. Because the LLM evaluators occasionally return ill-formed or nonsensical outputs, we re-query the model until a valid judgment is obtained. We report the results of FiRE with DeepSeek-R1 in the following sections. An ablation study on the choice of LLM backbone is detailed in Section 5.1. Detailed information for adopted LLMs are displayed in Appendix A and instructions for FiRE evaluators are provided in Appendix B.

### 4.3 METRICS

In our experiments, percentage agreement between various evaluators and human annotators is employed to showcase their performance on the proposed criterion-based pairwise evaluation. Position consistency and fairness are used to assess the position bias of LLM evaluators.

Table 2: Percentage agreement between model evaluators and human annotations on ranked overall pairwise data in EN→ZH and RU→ZH. Values are percentages (%); **Bold** indicates the best performance per language direction.

| | EN→ZH | RU→ZH |
|---|---|---|
| *Regression-Based* | | |
| KIWI-XL | 60.4 | 58.2 |
| KIWI-XXL | 61.4 | 61.2 |
| XCOMET-XL | 56.5 | 57.4 |
| XCOMET-XXL | 55.7 | 58.0 |
| MetricX-24-XXL | 61.6 | 67.1 |
| *Ranking-Based* | | |
| MT-Ranker-Base | 60.2 | 54.7 |
| MT-Ranker-Large | 61.0 | 60.9 |
| MT-Ranker-XXL | 60.7 | 61.6 |
| DeepSeek-R1-Direct-Rank | 64.3 | 66.7 |
| DeepSeek-R1-FiRE | **65.3** | **70.1** |

**Percentage agreement** measures the percentage of cases where the LLM evaluator's judgment aligns with the majority vote of human annotators. A higher percentage agreement indicates better alignment between LLM and human annotators, reflecting the model's ability to capture human preferences and evaluate the translation quality with specified criteria.

**Position consistency** is employed to evaluate the presence of position bias in our LLM evaluators. This metric measures how often the LLM evaluator makes the same judgment to a translation pair when their order is swapped. In simpler terms, imagine showing the LLM two translations, $y_1$ and $y_2$, and then showing the same translations again, but this time with $y_2$ first and $y_1$ second. Position consistency checks if the LLM gives the same judgment both times. It is calculated as follows:

$$\frac{1}{N} \sum \mathbb{I}(\mathcal{M}(p, x, y_1, y_2) = \mathcal{M}(p, x, y_2, y_1)) \tag{1}$$

**Position fairness** assesses the potential positional preferences of LLM evaluators. Specifically, after combining the data before and after swapping the translation order, it calculates the distribution of choices for each LLM evaluator. A higher value for a particular choice indicates a stronger preference by the model for that option.

### 4.4 RESULTS IN OVERALL PAIRWISE EVALUATION

We evaluate competing MT evaluation methods in the standard overall pairwise ranking setting, where the goal is to decide which of two translations is better overall. Performance is measured as the percentage agreement between an evaluator's decision and human annotations on our ranked pairwise benchmark. Results for EN→ZH and RU→ZH are reported in Table 2.

For LLM evaluators, we report two variants: DeepSeek-R1-Direct-Rank, which elicits a single overall judgment; and DeepSeek-R1-FiRE, which aggregates the fine-grained judgments on faithfulness, fluency, and consistency of style into an overall decision. For each example, the fine-grained judgments are encoded as a triple $\{c_1, c_2, c_3\}$, where $c_1, c_2, c_3 \in \{A, B, E\}$ denote the preferred translation (A or B) or a tie (E) for faithfulness, fluency, and consistency of style, respectively. We first compare how many of the three criteria favor translation A versus translation B; if A is favored more often than B, the FiRE outcome is A, and vice versa. In case of a tie, we break ties lexicographically by the criteria order—faithfulness, then fluency, then consistency of style—by selecting the first judgment that is not E as the FiRE outcome. If three criteria are ties, FiRE produces E as overall.

LLM evaluators, particularly FiRE, align with human preferences substantially better than established regression-based and existing ranking-based methods. DeepSeek-R1-Direct-Rank attains the agreement on EN→ZH at 64.3%, and DeepSeek-R1-FiRE attains the best agreement (65.3% EN→ZH, 70.1% RU→ZH). These results exceed strong regression baselines such as MetricX-24-XXL (61.6% EN→ZH, 67.1% RU→ZH) and KIWI-XXL (61.4% EN→ZH, 61.2% RU→ZH), as well as a dedicated ranking-based evaluator, MT-Ranker-XXL (60.7% EN→ZH, 61.6% RU→ZH), indicating LLMs outperform trained metrics in ranking evaluation. To our knowledge, we are the first to report the performance of LLMs in ranking evaluation. Moreover, FiRE aggregates the fine-grained, criterion-specific judgments into a single overall decision. This aggregation enables it to exploit complementary evidence across faithfulness, fluency, and consistency of style, to smooth out near-

Table 3: Percentage agreement between model evaluators and human annotations on ranked pairwise data across different criteria. Values are percentages (%); **Bold** indicates the best performance per criterion and language direction.

| | Faithfulness | Fluency | Cons. of Style | Overall | Faithfulness | Fluency | Cons. of Style | Overall |
|---|---|---|---|---|---|---|---|---|
| | EN→ZH | | | | RU→ZH | | | |
| *Error-Based* | | | | | | | | |
| M-MAD | 45.9 | 25.2 | 19.3 | 43.6 | 55.4 | 24.9 | 17.5 | 51.9 |
| GEMBA-MQM | 37.9 | 32.9 | 3.0 | 41.5 | 39.8 | 29.9 | 5.4 | 37.6 |
| *Ranking-Based* | | | | | | | | |
| FiRE | **64.8** | **68.7** | **61.4** | **65.3** | **72.5** | **77.9** | **66.3** | **70.1** |

tie noise, and to better reflect human trade-offs among these criteria. As a result, FiRE outperforms both strong metric baselines and LLM evaluators used with direct overall-ranking prompts.

### 4.5 RESULTS IN FINE-GRAINED RANKING EVALUATION

We compare the performance of different evaluation paradigms when assessing translations based on specific quality criteria: faithfulness, fluency, and consistency of style. According to Table 3, a key finding is the consistent and substantial outperformance of our proposed fine-grained ranking evaluation over error-based methods across all criteria. Concretely, error-based metrics such as M-MAD and GEMBA-MQM reach only moderate agreement with human judgments on faithfulness and overall quality (around 38–55%), and their agreement drops on fluency and especially consistency of style (down to single-digit or low levels), while FiRE attains 64.8–72.5% agreement on faithfulness, 68.7–77.9% on fluency, and 61.4–66.3% on consistency of style, yielding the strongest overall agreements (65.3% for EN→ZH, 70.1% for RU→ZH). This performance gap stems from fundamental differences in their evaluation mechanisms. Error-based evaluators, relying on predefined error taxonomies and aggregation, excel at error diagnosis but fail to capture the nuanced differences between two translations. Fluency, for instance, transcends mere grammatical correctness to include naturalness and readability, while consistency of style involves subtleties of tone and register not easily captured by discrete error counts. In contrast, FiRE makes direct, criterion-guided comparative judgments, which allows it to leverage its extensive linguistic knowledge for a more nuanced assessment, evaluating how well each translation embodies the desired quality in its entirety.

### 4.6 INTER-SYSTEM COMPARISON

Building upon its demonstrated high alignment with human preferences in both overall (Section 4.4) and fine-grained (Section 4.5) ranking evaluations, FiRE can be effectively applied to conduct comprehensive system-level analyses. A significant advantage of FiRE is its ability to move beyond a single overall ranking score by providing a fine-grained breakdown of MT system performance across multiple quality dimensions. This multi-faceted evaluation offers deeper diagnostic insights into the specific strengths and weaknesses of each system. Details for Inter-system evaluation method are provided in Appendix D.

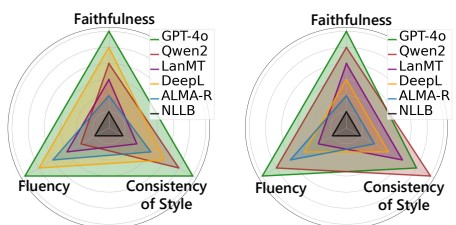

Figure 3: Fine-grained ranking of six MT systems based on all pairwise data in EN→ZH (left) and RU→ZH (right).

Figure 3 visually represents this fine-grained ranking for the six MT systems evaluated, using all pairwise data. These radar charts clearly illustrate how systems may vary in their performance across faithfulness, fluency, and consistency of style. For example, FiRE's analysis reveals that translations from ALMA-R, while exhibiting relatively high fluency, tend to score lower on faithfulness. Conversely, Qwen2 demonstrates notably strong stylistic consistency. Such nuanced distinctions are critical for understanding system behavior. Furthermore, the comparison across translation

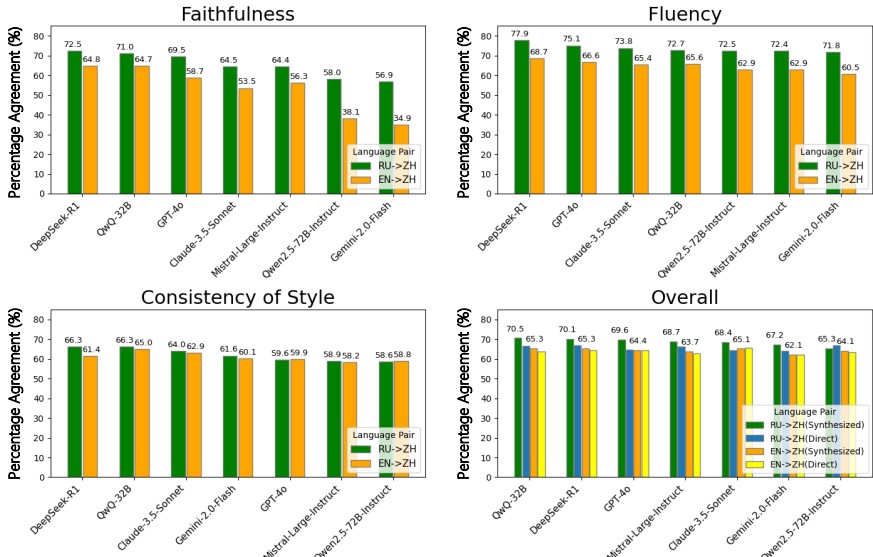

Figure 4: Percentage agreement of our proposed FiRE with various LLMs.

directions highlights differing system strengths; for instance, DeepL shows stronger performance in EN→ZH, whereas Qwen2 excels in RU→ZH.

This fine-grained information is a crucial complement to traditional overall system rankings (detailed overall rankings are presented in Table 16 in Appendix). The criterion-specific rankings provided by FiRE help explain why a system achieves a certain overall rank, offering actionable insights for system developers. Importantly, the overall system rankings derived from FiRE generally correspond to an aggregation of these fine-grained assessments, which further validates the internal consistency and effectiveness of our proposed method. Thus, FiRE not only determines which system is better but also elucidates how and in what aspects it excels or falls short, paving the way for more targeted MT system development and a more complete understanding of translation quality.

## 5 ANALYSIS

### 5.1 ABLATION STUDY ON USING DIFFERENT LLMS

Figure 4 summarizes FiRE's performance across LLM backbones. Most evaluators achieve comparable accuracy on fluency, consistency of style, and overall ranking; the main divergence appears on faithfulness, where Qwen2.5-72B-Instruct and Gemini-2.0-Flash lag behind. Across directions, RU→ZH is slightly stronger than EN→ZH. Reasoning-oriented judges (QwQ-32B and DeepSeek-R1) are more robust across criteria, suggesting that explicit reasoning is an important driver of accurate, stable judgments.

### 5.2 IMPACT OF DATA DIFFICULTY

To investigate the robustness of LLM evaluators, we analyzed their performance on subsets of our benchmark stratified by difficulty. Easy cases are defined as those where all three human annotators reached a consensus, while hard cases represent instances with agreement from only two out of three annotators, indicating more subtle distinctions. As shown in Table 4, LLM evaluator performance on EN→ZH ranked data stratified by difficulty (Easy vs. Hard) reveals a consistent decline in agreement with human annotations on harder examples across all criteria. For instance, the average performance drop from easy to hard on EN→ZH ranked data is 14.2% for faithfulness, 11.6% for fluency, and 6.5% for consistency of style. The comprehensive results in EN→ZH and RU→ZH , showcasing consistent trending, are displayed in Table 17 in the Appendix.

Table 4: Percentage agreement between LLM evaluators and human annotations on ranked pairwise data in EN→ZH. Values are percentages (%).

| | Faithfulness | | Fluency | | Cons. of Style | |
|---|---|---|---|---|---|---|
| | Easy | Hard | Easy | Hard | Easy | Hard |
| Qwen2.5-72B-Instruct | 43.3 | 26.6 | 67.1 | 53.5 | 62.0 | 55.0 |
| Mistral-Large-Instruct | 60.6 | 47.3 | 65.9 | 56.1 | 60.8 | 55.0 |
| GPT-4o | 63.4 | 49.1 | 70.2 | 58.5 | 62.7 | 56.4 |
| Claude-3.5-Sonnet | 58.2 | 44.1 | 69.0 | 57.1 | 66.3 | 58.8 |
| Gemini-2.0-Flash | 39.8 | 24.9 | 63.4 | 54.2 | 63.1 | 56.4 |
| DeepSeek-R1 | 68.6 | 57.1 | 73.2 | 58.5 | 64.3 | 57.8 |
| QwQ-32B | 69.6 | 54.7 | 68.8 | 58.5 | 67.8 | 61.6 |
| Average | 57.6 | 43.4 | 68.2 | 56.6 | 63.8 | 57.3 |

Table 5: Position bias of LLM evaluators indicated by position consistency and position fairness. Values are percentages (%). The results of fairness are percentage choices for A/B/E.

| | Faithfulness | | Fluency | | Consistency of Style | |
|---|---|---|---|---|---|---|
| | Consistency | Fairness | Consistency | Fairness | Consistency | Fairness |
| Qwen2.5-72B-Instruct | 65.5 | 20.0 / 27.5 / 52.5 | 61.5 | 32.2 / 62.0 / 5.8 | 59.5 | 32.8 / 63.8 / 3.5 |
| Mistral-Large-Instruct | 61.0 | 39.8 / 30.5 / 29.8 | 65.0 | 45.8 / 42.7 / 11.5 | 71.0 | 49.3 / 49.5 / 1.3 |
| GPT-4o | 58.5 | 55.0 / 28.8 / 16.3 | 71.5 | 57.5 / 41.5 / 1.0 | 51.5 | 70.0 / 29.5 / 1.0 |
| Claude-3.5-Sonnet | 63.5 | 33.3 / 32.7 / 34.0 | 67.5 | 39.5 / 52.2 / 8.3 | 70.5 | 57.5 / 41.5 / 1.0 |
| Gemini-2.0-Flash | 72.0 | 17.3 / 18.0 / 64.8 | 55.0 | 32.8 / 55.0 / 12.2 | 59.0 | 42.8 / 52.3 / 5.0 |
| DeepSeek-R1 | **73.5** | 44.3 / 40.3 / 15.5 | **81.0** | 47.8 / 49.2 / 3.0 | 71.0 | 42.8 / 53.0 / 4.3 |
| QwQ-32B | 65.5 | 49.3 / 37.0 / 13.8 | 70.0 | 39.8 / 58.3 / 2.0 | **76.5** | 50.8 / 49.2 / 0.0 |

It's important to note that even large models like GPT-4o do not exhibit complete immunity to the challenges posed by increased data difficulty, despite their enhanced contextual understanding abilities and multilingual capabilities. This observation suggests that factors beyond sheer model scale contribute to robust evaluation performance, indicating the need for a multifaceted and sophisticated approach to improve LLM-based MT evaluators for criterion-based pairwise evaluation.

## 5.3 BIAS OF LLM EVALUATORS

Apart from the performance of LLM evaluators, we investigate their bias in terms of position and self-enhancement. **Position bias** is the propensity of LLM evaluators to favor responses in certain positions within the prompt (Park et al., 2024). **Self-enhancement bias** refers to the tendency of LLM evaluators to exhibit a preference for responses generated by themselves (Ye et al., 2024).

**Position Bias.** As described in Section 4.3, we assess the position bias in LLM evaluators from two key aspects: position consistency and position fairness. We take EN→ZH pairwise data as an instance and present the position consistency and fairness of evaluators in Table 5. The average position consistency across all LLM evaluators is 65.6% for faithfulness, 67.4% for fluency, and 65.6% for consistency of style, suggesting that more than 30% of LLM judgments are altered after simply swapping the order of two translation candidates. The degree of position bias varies across LLMs and criteria. For example, GPT-4o is relatively robust on fluency but less stable on faithfulness and consistency of style, whereas Gemini-2.0-Flash attains stronger position consistency on faithfulness but exhibits noticeably weaker robustness on the other two criteria. Notably, models recognized for strong reasoning capabilities, such as DeepSeek-R1 and QwQ-32B, demonstrate higher position consistency. This robustness may be attributed to their training for complex reasoning, fostering meticulous instruction adherence, a deeper focus on intrinsic semantic and logical properties over superficial cues like order, or a more systematic, order-agnostic comparison process.

**Self-Enhancement Bias.** To investigate this, we analyze scenarios where translations generated by specific model versions (gpt-4o-2024-08-06 and Qwen2-72B-Instruct) were among the candidates, and these are then evaluated by slightly later versions from the same model series (gpt-4o-2024-11-20 and Qwen2.5-72B-Instruct, respectively). This setup allows us to examine if evaluators favor outputs from their own lineage. We compare the proportion of times the evaluator favored its own series' output against the preferences shown by other LLM evaluators and human annotators for same outputs. As illustrated in Figure 5, both Qwen2.5-72B-Instruct and GPT-4o exhibit a strong tendency for their self-generated translations and deviate from human favoritism across all specified

Table 6: Percentage agreement between model evaluators and human annotations on ranked pairwise data across different criteria. Values are percentages (%); **Bold** indicates the best performance per criterion and language direction. –indicates the model cannot produce criterion-based ranking.

| | Faithfulness | Fluency | Cons. of Style | Overall | Faithfulness | Fluency | Cons. of Style | Overall |
|---|---|---|---|---|---|---|---|---|
| | | JA→ZH | | | | HE→EN | | |
| *Regression-Based* | | | | | | | | |
| KIWI-XL | – | – | – | 65.3 | – | – | – | 67.1 |
| KIWI-XXL | – | – | – | 66.7 | – | – | – | 69.0 |
| XCOMET-XL | – | – | – | 60.7 | – | – | – | 62.5 |
| XCOMET-XXL | – | – | – | 61.5 | – | – | – | 64.8 |
| MetricX-24-XXL | – | – | – | 68.2 | – | – | – | 68.2 |
| *Error-Based* | | | | | | | | |
| M-MAD | 55.6 | 35.9 | 30.9 | 56.3 | 55.4 | 24.9 | 17.5 | 51.9 |
| GEMBA-MQM | 44.3 | 34.7 | 11.0 | 44.8 | 39.8 | 29.9 | 5.4 | 37.6 |
| *Ranking-Based* | | | | | | | | |
| MT-Ranker-Base | – | – | – | 60.8 | – | – | – | 60.8 |
| MT-Ranker-Large | – | – | – | 65.4 | – | – | – | 65.0 |
| MT-Ranker-XXL | – | – | – | 67.3 | – | – | – | 67.3 |
| QwQ-FiRE | **71.0** | **60.1** | **51.5** | **72.4** | **70.8** | **52.5** | **48.6** | **69.6** |

preferences in EN→ZH and RU→ZH. These LLM evaluators may have been extensively exposed to their own generated text during the post-training stage, particularly reinforcement learning, leading to an inherent preference for their own outputs (Chiang et al., 2024). This training mechanism makes the models more familiar with their own generation patterns, causing them to favor translations that align with their intrinsic probabilities instead of specified preferences during pairwise evaluation.

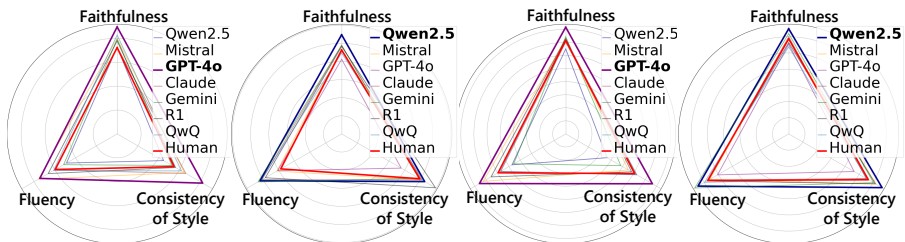

Figure 5: Self-enhancement bias of GPT-4o and Qwen2.5 in EN→ZH (left) and RU→ZH (right). Lines extending further outwards indicate stronger favoritism. Bolded line shows the preference of the respective evaluator for translations from its own model series. Human favoritism is depicted in red.

## 5.4 MQM DATASETS AND LOW-RESOURCE LANGUAGE

To connect our study with existing benchmarks and to assess performance on low-resource language pairs, we conduct experiments on MQM23 Hebrew-to-English (HE→EN) and MQM24 Japanese-to-Chinese (JA→ZH) (Freitag et al., 2021). As described in Section 4.1, we map MQM error tags to three criteria (faithfulness, fluency, consistency of style) following the grouping outlined in Table 13, and construct the corresponding synthesized pairwise dataset. The results in Table 6 show that FiRE achieves robust performance on the MQM datasets, including the low-resource language setting.

## 6 CONCLUSION

This paper introduced FiRE, a novel fine-grained ranking framework for criterion-based pairwise MT evaluation, addressing the need for more interpretable and human-aligned reference-free methods. We also presented the first human-annotated benchmark for this task, confirming the reliability of fine-grained human judgments. FiRE demonstrably outperforms existing methods in aligning with human preferences on this benchmark, offering richer insights across three criteria-faithfulness, fluency, and consistency of style. Our analysis of LLM evaluator biases and their handling of tied cases provides crucial guidance for developing more nuanced and reliable MT evaluation.

## ETHICS STATEMENT

We adhered to the ICLR Code of Ethics. Human annotators were recruited and fairly compensated, with fairness and respect ensured throughout the process. The annotators' proficiency is presented in Table 8. We built a labeled dataset based on source texts from WMT23, WMT24pp, with translations generated by language models and all sources properly cited.

## REPRODUCIBILITY STATEMENT

We provided the details for the data collection in Section 3.2 and the experiment setup in Section 4 and Appendix F. We will release our data after the review process is completed.

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

# A    DETAILS OF ADOPTED LLMS

Table 7 displays the adopted model versions, whether it is open-sourced or not, and the parameter sizes.

Table 7: Details regarding LLM evaluators in our experiments.

| Model | Version | Open-source | Parameters |
|---|---|---|---|
| **QwQ-32B** | QwQ-32B | ✔ | 32B |
| **Qwen2.5-72B-Instruct** | Qwen2.5-72B-Instruct | ✔ | 72B |
| **Mistral-Large-Instruct** | Mistral-Large-Instruct-2411 | ✔ | 123B |
| **DeepSeek-R1** | DeepSeek-R1 | ✔ | 671B |
| **GPT-4o** | gpt-4o-2024-11-20 | ✘ | N/A |
| **Claude-3.5-Sonnet** | claude-3-5-sonnet-20241022 | ✘ | N/A |
| **Gemini-2.0-Flash** | gemini-2.0-flash | ✘ | N/A |

# B    PROMPTS

## B.1    FAITHFULNESS

---

**Prompt: Faithfulness**

You are a translation evaluator. Given a triple ([source], [A], [B]), where [A] and [B] are two translation candidates. Please compare two translation candidates based on the source language text under the Given Preference with Note and making a relative evaluation of their quality. Please answer based on the analysis and write the analysis and result in the format "analysis": "Accuracy of Information":..., "Accuracy of Named Entities":..., "result":....
The marked options are divided into three categories, with the following specific meanings:
  - A: The quality of [A] is higher than the quality of [B]
  - B: The quality of [B] is higher than the quality of [A]
  - E: The quality of [A] is equivalent to the quality of [B], and it is impossible to distinguish the superiority or inferiority
If both translations contain errors, please determine which translation has more significant errors (choose A or B), or if both have errors of similar severity (choose E).

### Preference ###
Faithfulness in terms of the following aspects:
  1. Accuracy of Information: faithful to the original text, with no missing, incorrect, or added information.
  2. Accuracy of Named Entities: names of people, places, organizations, and specialized terms, as well as times, quantities, currency, ratios, and other specifics that are accurately translated.

### Translation Evaluation ###
Source: *source text*
Translation A: *<translation A>*
Translation B: *<translation B>*
Make pairwise evaluation with the specified preference according to previous instructions.

---

## B.2 FLUENCY

---

**Prompt: Fluency**

You are a translation evaluator. Given a triple ([source], [A], [B]), where [A] and [B] are two translation candidates. Please compare two translation candidates based on the source language text under the Given Preference with Note and making a relative evaluation of their quality. Please answer based on the analysis and write the analysis and result in the format "analysis": "Lexical Quality":..., "Syntactic Quality":..., "Punctuation":..., "Untranslated":..., "result":....

The marked options are divided into three categories, with the following specific meanings:
 - A: The quality of [A] is higher than the quality of [B]
 - B: The quality of [B] is higher than the quality of [A]
 - E: The quality of [A] is equivalent to the quality of [B], and it is impossible to distinguish the superiority or inferiority

If both translations contain errors, please determine which translation has more significant errors (choose A or B), or if both have errors of similar severity (choose E).

### Preference ###
Fluency in terms of the following aspects:
 1. Lexical Quality: Proper word choice, parts of speech, spelling, and capitalization.
 2. Syntactic Quality: Correct sentence structure, word order.
 3. Punctuation: Punctuation incorrect according to target language conventions. Missing mark from a set of paired punctuation marks, such as a missing parenthesis or quote mark.
 4. Untranslated: untranslated names of people or places.

### Translation Evaluation ###
Source: *<source text>*
Translation A: *<translation A>*
Translation B: *<translation B>*
Make pairwise evaluation with the specified preference according to previous instructions.

---

## B.3 CONSISTENCY OF STYLE

---

**Prompt: Consistency of Style**

You are a translation evaluator. Given a triple ([source], [A], [B]), where [A] and [B] are two translation candidates. Please compare two translation candidates based on the source language text under the Given Preference with Note and making a relative evaluation of their quality. Please answer based on the analysis and write the analysis and result in the format {"analysis": {"Tone Matching":..., "Emotional Preservation":..., "Writing Style":...}, "result":...}.

The marked options are divided into three categories, with the following specific meanings:
 - A: The quality of [A] is higher than the quality of [B]
 - B: The quality of [B] is higher than the quality of [A]
 - E: The quality of [A] is equivalent to the quality of [B], and it is impossible to distinguish the superiority or inferiority

If both translations contain errors, please determine which translation has more significant errors (choose A or B), or if both have errors of similar severity (choose E).

### Preference ###
Consistency of Style in terms of the following aspects:
 1. Tone Matching: The translated text's tone should match the source, whether academic, technical, or conversational.
 2. Emotional Preservation: The translation should convey the original text's emotional tone or mood, whether positive, negative or neutral. The translation should maintain the original mood, whether polite, assertive, or anger...

---

3. Writing Style: The translation should reflect the original style, whether concise and direct or detailed and thorough.

### Translation Evaluation ###
Source: *<source text>*
Translation A: *<translation A>*
Translation B: *<translation B>*
Make pairwise evaluation with the specified preference according to previous instructions.

## B.4  OVERALL

**Prompt: Overall**

You are a translation evaluator. Given a triple ([source], [A], [B]), where [A] and [B] are two translation candidates. Please compare two translation candidates based on the source language text and making a relative evaluation of their quality. Please answer based on analysis and write the analysis and result in the format {"analysis": ..., "result":...}.
The marked options are divided into three categories, with the following specific meanings:
   - A: The quality of [A] is higher than the quality of [B]
   - B: The quality of [B] is higher than the quality of [A]
   - E: The quality of [A] is equivalent to the quality of [B], and it is impossible to distinguish the superiority or inferiority
If both translations contain errors, please determine which translation has more significant errors (choose A or B), or if both have errors of similar severity (choose E).

### Translation Evaluation ###
Source: *source text*
Translation A: *<translation A>*
Translation B: *<translation B>*
Make pairwise evaluation with the specified preference according to previous instructions.

## C  DETAILS OF ANNOTATION GUIDELINES

The annotation guidelines consist of two main components. The first provides an overview of the annotation task, including the input format and the overall objective. The second specifies the operational definitions of the three evaluation criteria, accompanied by illustrative examples.

## D  SYSTEM-LEVEL EVALUATION METHOD

We estimate system-level performance using a normalized Copeland scoring rule. For each system $i$, we collect all pairwise comparisons involving $i$. In a comparison between systems $i$ and $j$, system $i$ receives $+1$ point if its translation is preferred, $+0.5$ points if the two translations are judged tied, and $0$ otherwise. Formally, let $\text{Wins}_{ij}$ denote the number of times system $i$ is preferred over system $j$, $\text{Ties}_{ij}$ the number of ties between $i$ and $j$, and $\text{Matches}_{ij}$ the total number of pairwise comparisons between $i$ and $j$. The score for system $i$ is

$$\text{Score}i = \frac{\sum_{j \neq i}\left(\text{Wins}_{ij} + 0.5\text{Ties}_{ij}\right)}{\sum_{j \neq i}\text{Matches}_{ij}} \tag{2}$$

The final score for system $i$ is the average of these points across all comparisons involving $i$, and systems are ranked by sorting these scores in descending order. This normalization accommodates unbalanced designs where different pairs $(i, j)$ may have different numbers of comparisons.

Table 8: Annotator's qualification based on CEFR[1] proficiency levels.

| Annotator | EN→ZH 1 | EN→ZH 2 | EN→ZH 3 | RU→ZH 1 | RU→ZH 2 | RU→ZH 3 |
|---|---|---|---|---|---|---|
| English | C1 | C1 | C1 | - | - | - |
| Russian | - | - | - | B2 | B1 | B2 |
| Chinese | C2 | C2 | C2 | C2 | C2 | C2 |

## E  PROFICIENCY OF ANNOTATORS

We followed the Common European Framework of Reference for Languages (CEFR) (Council of Europe), a guideline used to describe the achievements of learners of foreign languages across Europe and in other countries, and listed the proficiency of annotators in Table 8. Six levels in the CEFR are described as follows:

- **A1 (Breakthrough)**: Can understand and use familiar everyday expressions and very basic phrases aimed at the satisfaction of needs of a concrete type. Can introduce him/herself and others and can ask and answer questions about personal details such as where he/she lives, people he/she knows and things he/she has. Can interact in a simple way provided the other person talks slowly and clearly and is prepared to help.

- **A2 (Waystage)**: Can understand sentences and frequently used expressions related to areas of most immediate relevance (e.g. very basic personal and family information, shopping, local geography, employment). Can communicate in simple and routine tasks requiring a simple and direct exchange of information on familiar and routine matters. Can describe in simple terms aspects of his/her background, immediate environment and matters in areas of immediate need.

- **B1 (Threshold)**: Can understand the main points of clear standard input on familiar matters regularly encountered in work, school, leisure, etc. Can deal with most situations likely to arise whilst travelling in an area where the language is spoken. Can produce simple connected text on topics which are familiar or of personal interest. Can describe experiences and events, dreams, hopes  ambitions and briefly give reasons and explanations for opinions and plans.

- **B2 (Vantage)**: Can understand the main ideas of complex text on both concrete and abstract topics, including technical discussions in his/her field of specialisation. Can interact with a degree of fluency and spontaneity that makes regular interaction with native speakers quite possible without strain for either party. Can produce clear, detailed text on a wide range of subjects and explain a viewpoint on a topical issue giving the advantages and disadvantages of various options.

- **C1 (Advanced)**: Can understand a wide range of demanding, longer texts, and recognise implicit meaning. Can express him/herself fluently and spontaneously without much obvious searching for expressions. Can use language flexibly and effectively for social, academic and professional purposes. Can produce clear, well-structured, detailed text on complex subjects, showing controlled use of organisational patterns, connectors and cohesive devices.

- **C2 (Mastery)**: Can understand with ease virtually everything heard or read. Can summarise information from different spoken and written sources, reconstructing arguments and accounts in a coherent presentation. Can express him/herself spontaneously, very fluently and precisely, differentiating finer shades of meaning even in more complex situations.

## F  COMPUTATION SOURCE

All the experiments were done on NVIDIA A100 GPUs with 80G memory and CUDA 11.2, with driver 460.106.00. For MT-Ranker, we used a single GPU for the deployment of MT-Ranker-Base, a single GPU for MT-Ranker-Large respectively, and two GPUs for MT-Ranker-XXL.

Table 9: Percentage agreement between LLM evaluators and human annotations across different criteria in EN→ZH and RU→ZH. **Ranked** data points represent pairwise comparisons where one translation candidate is preferred ($y_1 \succ y_2$ or $y_2 \succ y_1$). **Tied** data points indicate equivalent candidates ($y_1 \sim y_2$). Values are percentages (%); **bold** indicates the best performance per criterion and language direction.

| | Faithfulness | | Fluency | | Consistency of Style | |
|---|---|---|---|---|---|---|
| | **Ranked** | **Tied** | **Ranked** | **Tied** | **Ranked** | **Tied** |
| | | | EN→ZH | | | |
| Qwen2.5-72B-Instruct | 38.1 | 66.4 | 62.9 | 12.5 | 58.8 | 5.1 |
| Mistral-Large-Instruct | 56.3 | 42.8 | 62.3 | 12.6 | 58.2 | 3.1 |
| GPT-4o | 58.7 | 36.5 | 66.6 | 2.2 | 59.9 | 2.8 |
| Claude-3.5-Sonnet | 53.5 | 45.5 | 65.4 | 11.5 | 62.9 | 2.5 |
| Gemini-2.0-Flash | 34.9 | 74.5 | 60.5 | 18.3 | 60.1 | 7.1 |
| DeepSeek-R1 | **64.8** | 22.0 | **68.7** | 6.3 | 61.4 | 2.9 |
| QwQ-32B | 64.7 | 21.8 | 65.6 | 4.0 | **65.0** | 1.0 |
| | | | RU→ZH | | | |
| Qwen2.5-72B-Instruct | 58.0 | 50.5 | 72.5 | 10.0 | 58.6 | 2.2 |
| Mistral-Large-Instruct | 64.4 | 33.2 | 72.4 | 7.1 | 58.9 | 1.5 |
| GPT-4o | 69.5 | 22.1 | 75.1 | 2.0 | 59.6 | 1.4 |
| Claude-3.5-Sonnet | 64.5 | 41.9 | 73.8 | 10.0 | 64.0 | 2.5 |
| Gemini-2.0-Flash | 56.9 | 63.9 | 71.8 | 12.5 | 61.6 | 4.1 |
| DeepSeek-R1 | **72.5** | 10.3 | **77.9** | 2.6 | **66.3** | 1.6 |
| QwQ-32B | 71.0 | 12.7 | 72.7 | 1.8 | **66.3** | 0.6 |

For LLM evaluators, we call the API of GPT-4o, Claude-3.5-Sonnet, Gemini-2.0-Flash, Mistral-Large-Instruct, and DeepSeek-R1. We use GPUs for the development of Qwen2.5-72B-Instruct, and QwQ-32B.

## G RANKED V.S. TIED

The results of ranked data versus tied data, shown in Table 9, reveal a critical challenge for LLMs in recognizing and evaluating translation pairs deemed equivalent by human annotators. This difficulty is particularly pronounced for fluency and consistency of style, as demonstrated by GPT-4o's extremely low 2.2% agreement on EN→ZH fluency-tied data. This suggests that LLMs struggle to discern subtle differences in translation quality when the options are very close, often resorting to making distinctions even when human annotators perceive parity. This tendency to over-discriminate could stem from the LLMs' training on large datasets where they are primarily tasked with identifying the best option, potentially hindering their ability to recognize and accept equally valid translations.

Future work should explore methods to better calibrate LLMs for recognizing equivalence, potentially through targeted training on tied pairs or by developing prompting strategies that explicitly encourage consideration of similarity. Our benchmark, by including distinct labels for ranked and tied comparisons, provides an essential resource for studying this phenomenon and driving progress in developing more nuanced evaluators.

## H THE USE OF LARGE LANGUAGE MODELS (LLMS)

We use LLMs to help us polish our paper.

## I    DETAILS OF TRANSLATION SYSTEMS

The details of the adopted MT systems during data collection are displayed in Table 10.

Table 10: Details of MT Systems. N/A indicates disclosed information.

| MT System | Version/Date | Open-source | Parameters | Architecture |
|---|---|---|---|---|
| NLLB-200-1.3B | HuggingFace | ✔ | 1.3B | Encoder-Decoder |
| ALMA-13B-R | HuggingFace | ✔ | 13B | Decoder-only |
| Qwen2-72B-Instruct | HuggingFace | ✔ | 72B | Decoder-only |
| GPT-4o | gpt-4o-2024-08-06 | ✘ | N/A | Decoder-only |
| DeepL | 2024-11-14 | ✘ | N/A | N/A |
| LanMT | 2024-11-20 | ✘ | N/A | N/A |

## J    THE USE OF NLLB-200-1.3B AND DOWNSAMPLING

Our goal in constructing the dataset was to cover a broad spectrum of translation quality, from relatively weak to very strong systems. We therefore deliberately included models of different scales and types, ranging from NLLB-200-1.3B through 13B and 72B LLMs to strong commercial systems. During pilot annotation, however, we observed that NLLB-200-1.3B produced substantially worse translations than the other systems, making many pairs involving this model extremely easy to judge (annotators almost always deemed it clearly inferior). Such trivial comparisons offer limited value for analyzing fine-grained human preferences and for differentiating strong evaluators. Consequently, we downsampled pairs involving NLLB-200-1.3B to prevent these easy cases from dominating the benchmark. This choice was not driven by hardware limitations—we could have used larger NLLB variants—but by annotation quality and dataset balance considerations. Our main findings are instead supported by comparisons among modern, strong systems (Qwen2-72B, GPT-4o, DeepL, LanMT, ALMA-13B-R), and are robust to the presence of weaker models.

## K    FUTURE DIRECTIONS

### K.1    SENTENCE-LEVEL V.S. PARAGRAPH-/DOCUMENT-LEVEL

Following most prior MT meta-evaluation benchmarks, we operate at the segment (sentence) level, where each data point consists of one source sentence and two translation candidates. While this design facilitates controlled comparison with existing metrics, recent studies have shown that paragraph- and document-level evaluation are important for machine translation (Deutsch et al., 2023; Sun et al., 2024b). The proposed framework, FiRE, itself is agnostic to segment length and can, in principle, be applied to paragraphs or documents. Therefore, extending FiRE to paragraph- and document-level evaluation and curating more fine-grained ranking evaluation benchmark at the paragraph-level or document-level are important directions for future work.

### K.2    POSITION BIAS OF HUMAN ANNOTATORS

Our analysis of position bias focuses on LLM evaluators because, unlike humans, they do not have long-term memory of previously seen samples, making position bias a more intrinsic modeling issue. In contrast, human annotators may implicitly remember sentences or earlier judgments during the annotation process, which could introduce confounding effects (e.g., recall or learning bias) when re-presenting the same pairs in reversed order. To avoid such potential contamination and ensure fair experimental conditions, we therefore did not conduct a detailed position consistency study on human annotators in this work. However, it could provide valuable information for the community if researchers can solve this dilemma and study the position bias of human annotators.

## L    ANNOTATED JAPANESE-TO-CHINESE DATASET FROM WMT24++

We additionally evaluate FiRE and other baselines on a JA→ZH test set from WMT24++, using the same annotation protocol described in Section 3.2. Results are displayed in Table 11. On this dataset, we again observe: 1) high inter-annotator agreement across the three criteria. 2) consistent results that FiRE outperforms strong baselines and remains robust across criteria.

Table 11: Percentage agreement between model evaluators and human annotations on ranked pair-wise data across different criteria in JA→ZH. Values are percentages (%); **Bold** indicates the best performance per criterion and language direction. † indicates Synthesized FiRE.

|  | Faithfulness | Fluency | Cons. of Style | Overall |
|---|---|---|---|---|
|  |  | JA→ZH |  |  |
| *Error-Based* |  |  |  |  |
| M-MAD | 61.9 | 39.6 | 28.3 | 60.2 |
| GEMBA-MQM | 46.7 | 41.6 | 10.5 | 46.9 |
| XCOMET-XL (MQM) | – | – | – | 28.6 |
| XCOMET-XXL (MQM) | – | – | – | 32.8 |
| *Regression-Based* |  |  |  |  |
| KIWI-XL | – | – | – | 66.9 |
| KIWI-XXL | – | – | – | 74.3 |
| XCOMET-XL | – | – | – | 65.3 |
| XCOMET-XXL | – | – | – | 65.9 |
| MetricX-24-XXL | – | – | – | 76.1 |
| *Ranking-Based* |  |  |  |  |
| MT-Ranker-Base | – | – | – | 65.0 |
| MT-Ranker-Large | – | – | – | 70.3 |
| MT-Ranker-XXL | – | – | – | 75.7 |
| Qwen2.5-72B-Instruct | 73.3 | 76.6 | 80.0 | 79.4 / 78.6† |
| Mistral-Large-Instruct | 81.0 | 80.3 | 84.9 | 84.0 / **84.7**† |
| GPT-4o | 80.6 | 80.8 | 80.2 | **85.1** / 80.5† |
| Claude-3.5-Sonnet | **81.6** | **82.5** | **85.3** | 83.7 / 83.7† |
| Gemini-2.0-Flash | 60.1 | 74.8 | 74.4 | 75.1 / 77.7† |
| DeepSeek-R1 | 78.9 | 81.0 | 78.1 | 81.2 / 82.5† |
| QwQ-32B | 78.9 | 80.1 | 77.3 | 80.9 / 81.9† |

## M    MAJORITY VOTE V.S. PER-ANNOTATOR COMPARISON

We follow the common practice in MT studies of using the majority vote as the gold label for meta-evaluation. Individual annotations might be noisy, and using the majority vote reduces variance and provides a more stable target, especially in our 3-annotator, 3-class setting. This is also consistent with how we compute Fleiss' kappa in Table 1, where the aggregated labels summarize substantial inter-annotator agreement. Table 12 displays the percentage agreement between LLM evaluators and each human annotator, indicating similar results and a consistent trend of majority vote.

Table 12: Percentage agreement between model evaluators and three human annotations (Annotator1 / Annotator2 / Annotator3) on ranked pairwise data across different criteria in EN→ZH and RU→ZH. Values are percentages (%).

| | Faithfulness | Fluency | Cons. of Style | Overall |
|---|---|---|---|---|
| | EN→ZH | | | |
| *Error-Based* | | | | |
| M-MAD | 45.6 / 44.8 / 46.4 | 23.8 / 24.5 / 26.0 | 19.3 / 19.7 / 19.4 | 43.6 / 42.0 / 44.9 |
| GEMBA-MQM | 38.7 / 39.5 / 37.5 | 31.2 / 30.5 / 33.7 | 2.5/ 3.3 / 2.7 | 41.9 / 41.3 / 41.3 |
| XCOMET-XL (MQM) | – | – | – | 57.6 / 55.6 / 57.5 |
| XCOMET-XXL (MQM) | – | – | – | 57.1 / 54.7 / 55.0 |
| *Regression-Based* | | | | |
| KIWI-XL | – | – | – | 61.3 / 60.0 / 61.3 |
| KIWI-XXL | – | – | – | 61.6 / 60.3 / 61.0 |
| XCOMET-XL | – | – | – | 57.6 / 55.6 / 57.5 |
| XCOMET-XXL | – | – | – | 57.1 / 54.7 / 55.0 |
| MetricX-24-XXL | – | – | – | 62.2 / 61.6 / 60.3 |
| *Ranking-Based* | | | | |
| MT-Ranker-Base | – | – | – | 60.1 / 58.8 / 59.0 |
| MT-Ranker-Large | – | – | – | 59.8 / 60.5 / 60.5 |
| MT-Ranker-XXL | – | – | – | 61.2 / 59.4 / 59.4 |
| Qwen2.5-72B-Instruct | 38.0 / 37.7 / 38.4 | 62.1 / 61.7 / 62.7 | 58.7 / 60.2 / 57.0 | 63.6 / 63.2 / 63.3 |
| Mistral-Large-Instruct | 53.9 / 55.1 / 55.4 | 60.5 / 59.8 / 63.9 | 57.1 / 55.6 / 59.9 | 63.0 / 62.2 / 63.6 |
| GPT-4o | 57.6 / 58.8 / 57.6 | 65.2 / 63.1 / 67.8 | 56.1 / 62.9 / 58.7 | 63.3 / 63.6 / 63.6 |
| Claude-3.5-Sonnet | 52.4 / 51.1 / 53.5 | 64.2 / 62.5 / 64.6 | 59.1 / 60.6 / 63.6 | 64.3 / 63.1 / 64.1 |
| Gemini-2.0-Flash | 33.4 / 34.1 / 35.0 | 58.6 / 56.9 / 62.3 | 55.9 / 59.7 / 60.1 | 61.1 / 60.0 / 61.9 |
| DeepSeek-R1 | 63.5 / 61.9 / 65.9 | 67.1 / 65.6 / 71.1 | 59.4 / 59.5 / 63.0 | 64.4 / 63.2 / 66.2 |
| QwQ-32B | 63.3 / 63.6 / 64.4 | 64.1 / 63.7 / 67.6 | 62.2 / 63.5 / 65.1 | 64.4 / 64.5 / 65.3 |
| | EN→ZH | | | |
| *Error-Based* | | | | |
| M-MAD | 56.4 / 54.5 / 54.1 | 24.3 / 23.6 / 26.0 | 16.3 / 17.6 / 16.2 | 52.1 / 51.3 / 51.2 |
| GEMBA-MQM | 45.4 / 45.3 / 43.4 | 30.0 / 29.5 / 29.9 | 4.9 / 5.9 / 5.5 | 42.1 / 43.0 / 42.0 |
| XCOMET-XL (MQM) | – | – | – | 57.3 / 57.0 / 58.7 |
| XCOMET-XXL (MQM) | – | – | – | 56.5 / 58.2 / 57.5 |
| *Regression-Based* | | | | |
| KIWI-XL | – | – | – | 57.7 / 58.2 / 57.8 |
| KIWI-XXL | – | – | – | 60.5 / 61.2 / 61.3 |
| XCOMET-XL | – | – | – | 57.3 / 57.0 / 58.7 |
| XCOMET-XXL | – | – | – | 56.5 / 58.2 / 57.5 |
| MetricX-24-XXL | – | – | – | 66.2 / 67.9 / 67.0 |
| *Ranking-Based* | | | | |
| MT-Ranker-Base | – | – | – | 54.3 / 54.4 / 55.8 |
| MT-Ranker-Large | – | – | – | 60.5 / 59.7 / 60.3 |
| MT-Ranker-XXL | – | – | – | 60.9 / 61.9 / 61.6 |
| Qwen2.5-72B-Instruct | 56.1 / 58.6 / 57.1 | 71.3 / 72.6 / 71.8 | 57.6 / 61.6 / 61.2 | 64.4 / 65.3 / 64.9 |
| Mistral-Large-Instruct | 64.6 / 64.0 / 63.9 | 71.3 / 72.2 / 72.6 | 59.0 / 58.4 / 61.2 | 67.2 / 68.0 / 68.4 |
| GPT-4o | 67.8 / 70.0 / 68.0 | 72.5 / 74.5 / 75.5 | 57.6 / 59.5 / 61.9 | 67.2 / 70.1 / 68.4 |
| Claude-3.5-Sonnet | 64.2 / 64.0 / 63.7 | 73.4 / 73.2 / 73.9 | 64.0 / 61.9 / 63.6 | 67.7 / 67.6 / 68.4 |
| Gemini-2.0-Flash | 55.3 / 55.2 / 55.9 | 72.5 / 72.0 / 71.0 | 60.4 / 65.4 / 62.9 | 66.7 / 66.6 / 67.7 |
| DeepSeek-R1 | 71.0 / 72.9 / 71.4 | 76.5 / 78.1 / 77.1 | 66.8 / 64.5 / 65.6 | 69.5 / 70.1 / 69.3 |
| QwQ-32B | 69.5 / 71.0 / 70.3 | 72.9 / 72.3 / 73.1 | 65.7 / 64.2 / 69.1 | 68.9 / 70.3 / 70.5 |

Table 13: Mapping between error types and three criteria.

| | **M-MAD** | **GEMBA-MQM** |
|---|---|---|
| Faithfulness | Accuracy, Terminology | Accuracy, Terminology, Non-translation |
| Fluency | Fluency | Fluency |
| Consistency of Style | Style | Style |

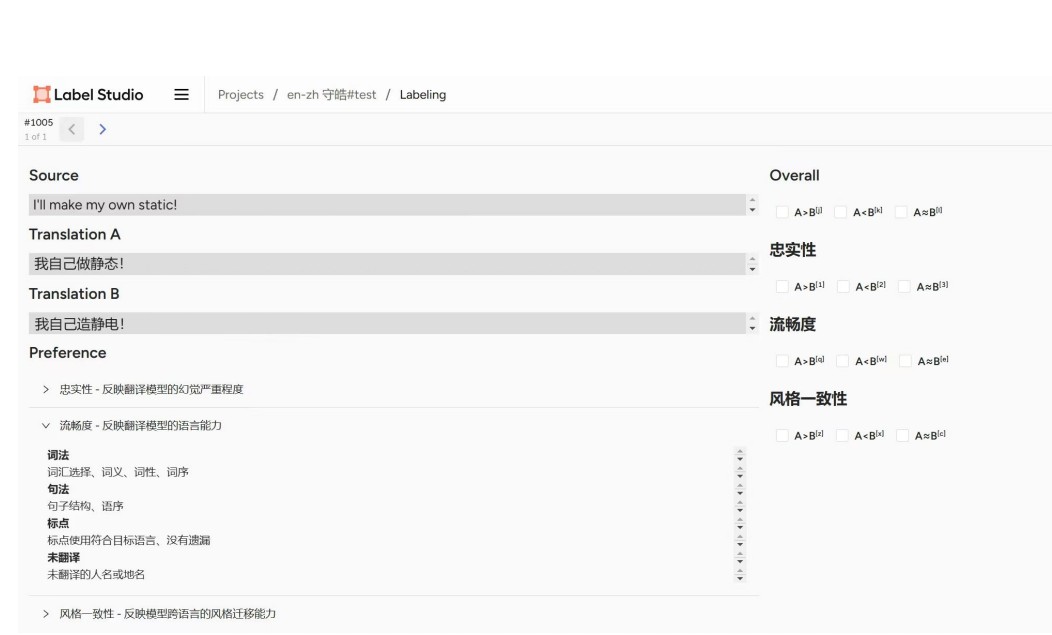

Figure 6: An example of annotation interface.

Table 14: Percentage agreement between model evaluators and human annotations on ranked pairwise data across different criteria in EN→ZH and RU→ZH. Values are percentages (%); **Bold** indicates the best performance per criterion and language direction. † indicates Synthesized FiRE.

| | Faithfulness | Fluency | Cons. of Style | Overall |
|---|---|---|---|---|
| | | EN→ZH | | |
| *Error-Based* | | | | |
| M-MAD | 45.9 | 25.2 | 19.3 | 43.6 |
| GEMBA-MQM | 37.9 | 32.9 | 3.0 | 41.5 |
| XCOMET-XL (MQM) | – | – | – | 38.7 |
| XCOMET-XXL (MQM) | – | – | – | 42.0 |
| *Regression-Based* | | | | |
| KIWI-XL | – | – | – | 60.4 |
| KIWI-XXL | – | – | – | 61.4 |
| XCOMET-XL | – | – | – | 56.5 |
| XCOMET-XXL | – | – | – | 55.7 |
| MetricX-24-XXL | – | – | – | 61.6 |
| *Ranking-Based* | | | | |
| MT-Ranker-Base | – | – | – | 60.2 |
| MT-Ranker-Large | – | – | – | 61.0 |
| MT-Ranker-XXL | – | – | – | 60.7 |
| Qwen2.5-72B-Instruct | 38.1 | 62.9 | 58.8 | 63.2 / 64.1$^\dagger$ |
| Mistral-Large-Instruct | 56.3 | 62.9 | 58.2 | 62.6 / 63.7$^\dagger$ |
| GPT-4o | 58.7 | 66.6 | 59.9 | **64.3** / 64.4$^\dagger$ |
| Claude-3.5-Sonnet | 53.5 | 65.4 | 62.9 | 65.5 / 65.1$^\dagger$ |
| Gemini-2.0-Flash | 34.9 | 60.5 | 60.1 | 62.2 / 62.1$^\dagger$ |
| DeepSeek-R1 | **64.8** | **68.7** | 61.4 | **64.3** / 65.3$^\dagger$ |
| QwQ-32B | 64.7 | 65.6 | **65.0** | 63.8 / 65.3$^\dagger$ |
| | | RU→ZH | | |
| *Error-Based* | | | | |
| M-MAD | 55.4 | 24.9 | 17.5 | 51.9 |
| GEMBA-MQM | 39.8 | 29.9 | 5.4 | 37.6 |
| XCOMET-XL (MQM) | – | – | – | 38.6 |
| XCOMET-XXL (MQM) | – | – | – | 40.0 |
| *Regression-Based* | | | | |
| KIWI-XL | – | – | – | 58.2 |
| KIWI-XXL | – | – | – | 61.2 |
| XCOMET-XL | – | – | – | 57.4 |
| XCOMET-XXL | – | – | – | 58.0 |
| MetricX-24-XXL | – | – | – | **67.1** |
| *Ranking-Based* | | | | |
| MT-Ranker-Base | – | – | – | 54.7 |
| MT-Ranker-Large | – | – | – | 60.9 |
| MT-Ranker-XXL | – | – | – | 61.6 |
| Qwen2.5-72B-Instruct | 58.0 | 72.5 | 58.6 | 67.0 / 65.3$^\dagger$ |
| Mistral-Large-Instruct | 64.4 | 72.4 | 58.9 | 66.2 / 68.7$^\dagger$ |
| GPT-4o | 69.5 | 75.1 | 59.6 | 64.6 / 69.6$^\dagger$ |
| Claude-3.5-Sonnet | 64.5 | 73.8 | 64.0 | 64.2 / 68.4$^\dagger$ |
| Gemini-2.0-Flash | 56.9 | 71.8 | 61.6 | 64.0 / 67.2$^\dagger$ |
| DeepSeek-R1 | **72.5** | **77.9** | **66.3** | 66.7 / 70.1$^\dagger$ |
| QwQ-32B | 71.0 | 72.7 | **66.3** | 66.4 / 70.5$^\dagger$ |

Table 15: Scores and Ranking of six MT systems calculated by FiRE.

| | Faithfulness | Fluency | Consistency of Style | Overall |
|---|---|---|---|---|
| | | EN→ZH | | |
| GPT-4o | 0.81 (1) | 0.70 (1) | 0.64 (1) | 0.72 (1) |
| DeepL | 0.61 (2) | 0.54 (2) | 0.54 (3) | 0.54 (2) |
| LanMT | 0.44 (4) | 0.44 (4) | 0.41 (5) | 0.39 (5) |
| Qwen2 | 0.49 (3) | 0.42 (5) | 0.55 (2) | 0.50 (3) |
| ALMA-R | 0.29 (5) | 0.52 (3) | 0.43 (4) | 0.44 (4) |
| NLLB | 0.15 (6) | 0.01 (6) | 0.04 (6) | 0.08 (6) |
| | | RU→ZH | | |
| GPT-4o | 0.81 (1) | 0.69 (1) | 0.61 (2) | 0.62 (1) |
| DeepL | 0.45 (4) | 0.42 (4) | 0.42 (4) | 0.47 (4) |
| LanMT | 0.49 (3) | 0.40 (5) | 0.53 (3) | 0.48 (3) |
| Qwen2 | 0.56 (2) | 0.61 (2) | 0.69 (1) | 0.61 (2) |
| ALMA-R | 0.33 (5) | 0.53 (3) | 0.38 (5) | 0.40 (5) |
| NLLB | 0.12 (6) | 0.06 (6) | 0.00 (6) | 0.09 (6) |

Table 16: Ranking of six MT systems calculated by QwQ-32B. The source data comes from WMT24pp (Deutsch et al., 2025), with 997 source texts per language. ALMA-R does not support JA→ZH and ZH→JA.

| | Faithfulness | Fluency | Consistency of Style | Overall | Faithfulness | Fluency | Consistency of Style | Overall |
|---|---|---|---|---|---|---|---|---|
| | | DE→EN | | | | | | |
| GPT-4o | 0.74 (1) | 0.59 (2) | 0.69 (1) | 0.66 (1) | 0.79 (1) | 0.64 (2) | 0.74 (1) | 0.71 (1) |
| DeepL | 0.55 (3) | 0.60 (1) | 0.62 (2) | 0.58 (3) | 0.61 (2) | 0.65 (1) | 0.66 (2) | 0.64 (2) |
| LanMT | 0.40 (5) | 0.36 (5) | 0.37 (5) | 0.38 (5) | 0.41 (5) | 0.35 (5) | 0.39 (5) | 0.38 (5) |
| Qwen2 | 0.58 (2) | 0.59 (3) | 0.55 (3) | 0.61 (2) | 0.50 (3) | 0.48 (4) | 0.49 (3) | 0.49 (4) |
| ALMA-R | 0.54 (4) | 0.55 (4) | 0.55 (4) | 0.53 (4) | 0.46 (4) | 0.56 (3) | 0.47 (4) | 0.52 (3) |
| NLLB | 0.19 (6) | 0.31 (6) | 0.22 (6) | 0.23 (6) | 0.23 (6) | 0.32 (6) | 0.25 (6) | 0.25 (6) |
| | | DE→ZH | | | | | ZH→DE | | |
| GPT-4o | 0.82 (1) | 0.65 (2) | 0. 78 (1) | 0.73 (1) | 0.86 (1) | 0.75 (1) | 0.81 (1) | 0.81 (1) |
| DeepL | 0.59 (3) | 0.64 (3) | 0.63 (3) | 0.62 (3) | 0.57 (2) | 0.60 (2) | 0.60 (2) | 0.60 (2) |
| LanMT | 0.37 (5) | 0.37 (5) | 0.37 (5) | 0.36 (5) | 0.40 (5) | 0.43 (5) | 0.40 (5) | 0.42 (5) |
| Qwen2 | 0.67 (2) | 0.67 (1) | 0.65 (2) | 0.70 (2) | 0.56 (3) | 0.60 (3) | 0.58 (3) | 0.59 (3) |
| ALMA-R | 0.40 (4) | 0.55 (4) | 0.43 (4) | 0.47 (4) | 0.49 (4) | 0.43 (4) | 0.46 (4) | 0.44 (4) |
| NLLB | 0.15 (6) | 0.12 (6) | 0.12 (6) | 0.13 (6) | 0.12 (6) | 0.19 (6) | 0.14 (6) | 0.15 (6) |
| | | JA→EN | | | | | EN→JA | | |
| GPT-4o | 0.84 (1) | 0.73 (1) | 0.80 (1) | 0.79 (1) | 0.83 (1) | 0.74 (1) | 0.72 (1) | 0.80 (1) |
| DeepL | 0.56 (3) | 0.57 (3) | 0.62 (3) | 0.58 (3) | 0.55 (3) | 0.64 (2) | 0.72 (2) | 0.58 (3) |
| LanMT | 0.37 (5) | 0.31 (5) | 0.33 (5) | 0.34 (5) | 0.50 (4) | 0.45 (4) | 0.41 (4) | 0.48 (4) |
| Qwen2 | 0.65 (2) | 0.69 (2) | 0.64 (2) | 0.69 (2) | 0.63 (2) | 0.63 (3) | 0.59 (3) | 0.65 (2) |
| ALMA-R | 0.41 (4) | 0.43 (4) | 0.43 (4) | 0.41 (4) | 0.33 (5) | 0.36 (5) | 0.28 (5) | 0.33 (5) |
| NLLB | 0.16 (6) | 0.27 (6) | 0.19 (6) | 0.20 (6) | 0.17 (6) | 0.19 (6) | 0.27 (6) | 0.17 (6) |
| | | JA→ZH | | | | | ZH→JA | | |
| GPT-4o | 0.83 (1) | 0.69 (2) | 0.77 (1) | 0.74 (2) | 0.84 (1) | 0.74 (1) | 0.73 (1) | 0.79 (1) |
| DeepL | 0.52 (3) | 0.57 (3) | 0.56 (3) | 0.54 (3) | 0.47 (3) | 0.55 (3) | 0.61 (2) | 0.49 (3) |
| LanMT | 0.36 (4) | 0.41 (4) | 0.40 (4) | 0.39 (4) | 0.44 (4) | 0.48 (4) | 0.46 (4) | 0.47 (4) |
| Qwen2 | 0.71 (2) | 0.73 (1) | 0.70 (2) | 0.75 (1) | 0.66 (2) | 0.64 (2) | 0.59 (3) | 0.67 (2) |
| ALMA-R | - | - | - | - | - | - | - | - |
| NLLB | 0.08 (5) | 0.10 (5) | 0.08 (5) | 0.08 (5) | 0.08 (5) | 0.09 (5) | 0.10 (5) | 0.09 (5) |

Table 17: Percentage agreement between LLM evaluators and human annotations across different criteria on ranked and tied pairwise data.

| | Ranked | | | | | | Tied | | | | | |
|---|---|---|---|---|---|---|---|---|---|---|---|---|
| | Faithfulness | | Fluency | | Cons. of Style | | Faithfulness | | Fluency | | Cons. of Style | |
| | Easy | Hard | Easy | Hard | Easy | Hard | Easy | Hard | Easy | Hard | Easy | Hard |
| EN→ZH | | | | | | | | | | | | |
| Qwen2.5-72B-Instruct | 43.3 | 26.6 | 67.1 | 53.5 | 62.0 | 55.0 | 70.2 | 57.7 | 13.4 | 10.8 | 5.4 | 3.9 |
| Mistral-Large-Instruct | 60.6 | 47.3 | 65.9 | 56.1 | 60.8 | 55.0 | 46.6 | 33.7 | 12.1 | 13.7 | 3.2 | 2.7 |
| GPT-4o | 63.4 | 49.1 | 70.2 | 58.5 | 62.7 | 56.4 | 41.1 | 25.8 | 2.8 | 1.0 | 3.3 | 1.2 |
| Claude-3.5-Sonnet | 58.2 | 44.1 | 69.0 | 57.1 | 66.3 | 58.8 | 49.5 | 36.2 | 12.1 | 10.3 | 2.8 | 1.2 |
| Gemini-2.0-Flash | 39.8 | 24.9 | 63.4 | 54.2 | 63.1 | 56.4 | 77.7 | 66.9 | 18.9 | 17.2 | 7.4 | 5.9 |
| DeepSeek-R1 | 68.6 | 57.1 | 73.2 | 58.5 | 64.3 | 57.8 | 25.4 | 14.1 | 7.6 | 3.9 | 3.5 | 1.0 |
| QwQ-32B | 69.6 | 54.7 | 68.8 | 58.5 | 67.8 | 61.6 | 23.3 | 18.4 | 4.3 | 3.4 | 0.9 | 0.8 |
| RU→ZH | | | | | | | | | | | | |
| Qwen2.5-72B-Instruct | 60.6 | 46.8 | 75.6 | 64.2 | 65.8 | 51.0 | 52.7 | 37.3 | 10.8 | 5.8 | 2.3 | 1.0 |
| Mistral-Large-Instruct | 67.1 | 53.2 | 75.2 | 64.8 | 64.5 | 53.1 | 35.0 | 23.0 | 7.7 | 4.3 | 1.2 | 3.5 |
| GPT-4o | 72.0 | 58.9 | 76.8 | 70.5 | 65.1 | 53.8 | 23.4 | 14.3 | 2.3 | 0.7 | 1.4 | 1.4 |
| Claude-3.5-Sonnet | 66.7 | 55.3 | 76.8 | 65.8 | 69.1 | 58.6 | 45.1 | 23.0 | 11.3 | 2.9 | 2.7 | 1.4 |
| Gemini-2.0-Flash | 59.2 | 47.5 | 75.8 | 61.1 | 69.1 | 53.8 | 66.8 | 46.8 | 13.6 | 6.4 | 4.2 | 3.5 |
| DeepSeek-R1 | 74.4 | 64.5 | 80.1 | 72.0 | 69.7 | 62.8 | 11.5 | 3.2 | 3.0 | 7.1 | 1.6 | 2.1 |
| QwQ-32B | 72.5 | 64.5 | 76.2 | 63.2 | 74.3 | 57.9 | 13.8 | 6.3 | 2.2 | 0.0 | 0.6 | 0.7 |

Table 18: Statistics of human-annotated dataset.

| | EN→ZH | | | | RU→ZH | | | |
|---|---|---|---|---|---|---|---|---|
| | Label | | Difficulty | | Label | | Difficulty | |
| | Ranked | Tied | Easy | Hard | Ranked | Tied | Easy | Hard |
| Faithfulness | 1029 | 545 | 1073 | 501 | 727 | 864 | 1324 | 267 |
| Fluency | 973 | 601 | 1069 | 505 | 710 | 882 | 1259 | 333 |
| Consistency of Style | 466 | 1102 | 1102 | 466 | 297 | 1297 | 1308 | 286 |
| Overall | 1317 | 257 | 1102 | 472 | 1239 | 349 | 1216 | 372 |

