# OpenReview forum: "FiRE: Fine-Grained Ranking Evaluation for Machine Translation"
_ICLR.cc/2026/Conference — Submitted to ICLR 2026_

### Official Review · Reviewer_Fo13 · 2025-10-28

**Soundness:** 3
**Presentation:** 3
**Contribution:** 3
**Rating:** 8
**Confidence:** 5

**Summary:**

The paper proposes fine-grained pair-wise ranking evaluation of machine translation using off-the-shelf LLMs. The evaluation is done on three complimentary dimensions: faithfulness, fluency, and consistency of style. The paper also provides the first human-annotated fine-grained machine translation evaluation benchmark.

**Strengths:**

1. Fine-grained human-annotated evaluation data was collected for meta-evaluation. The human annotations have high inter-rater agreement.
2. The paper provides comparison against strong baselines.
3. The paper studies the issue of position bias, showing strong position bias of LLMs and the need for position bias mitigation.
4. The paper also examines the LLM's preference for their own generations.
5. Performance on easy vs hard examples are shown, which validates that when human annotators find an example difficult the LLMs also struggle.

**Weaknesses:**

1. The paper only focuses on high resource languages and the author's collected data. It is understandable that collecting data for low resource languages is difficult. However as mentioned on table 10, existing MQM annotations could probably be directly mapped to the three evaluation criteria under consideration. An analysis on these MQM datasets would provide an independent verification of the proposed approach on independent datasets.
2. For position bias experiments, results have not been reported for each of the three evaluation criteria. Whether LLMs show more or less position bias on these different criteria would be an interesting research question.

**Questions:**

Was there any cases where the LLMs did not follow the prompt instruction and generated nonsensical answers? If so, how was it handled?

---

> ### Author Response · Authors · 2025-11-28
> **Response to Reviewer Fo13**
>
> Thank you for the very positive and detailed evaluation, and for highlighting our contributions on data collection, experiments, and analysis. Below we address the remaining weaknesses and the question in detail.
>
> ### 1. **Scope: low-resource languages and use of MQM datasets**
>
> > **Weakness 1:** The paper only focuses on high resource languages and the author's collected data. An analysis on these MQM datasets would provide an independent verification of the proposed approach on independent datasets.
> >
>
> We appreciate this suggestion and fully agree that testing on independent datasets and lower-resource scenarios is important. Following your recommendation, we have extended our experiments in two directions:
>
> - **MQM23 Hebrew→English (HE→EN) and MQM24 Japanese→Chinese (JA→ZH):** We conduct new experiments on the MQM23 HE→EN and MQM24 JA→ZH datasets. Based on the MQM taxonomy, we map fine-grained MQM error tags into our three criteria (faithfulness, fluency, and consistency of style) using the grouping outlined in Table 13, and then apply FiRE on this independent benchmark.
> - **Additional language pair:** We curated a new **Japanese→Chinese (JA→ZH)** evaluation set from WMT24++ with three human annotators following the same protocol as for EN→ZH and RU→ZH. On this dataset, we again observe: 1) high inter-annotator agreement across the three criteria. 2) consistent results that FiRE outperforms strong baselines and remains robust across criteria.
>
> In the revision, we updated as follows:
>
> - Add a subsection (**Section 5.4**) that introduces the MQM23 HE→EN and MQM24 JA→ZH experiments and summarizes the main findings.
> - Report the JA→ZH results in the **Appendix L** and **Table 11**, as a complementary language pair to EN→ZH and RU→ZH, and and highlight that our conclusions hold across these four directions (EN→ZH, RU→ZH, HE→EN, JA→ZH).
>
> We believe these additional experiments address the concern about reliance on our own collected data and strengthen the generality of our approach beyond high-resource language pairs.
>
> ### 2. Position bias per criterion
>
> > **Weakness 2:** Whether LLMs show more or less position bias on these different criteria would be an interesting research question.
> >
>
> Thank you for this helpful suggestion; we agree that criterion-specific position bias is both interesting and practically relevant.
>
> In the revision, we expand our analysis as follows:
>
> - For the **EN→ZH** setting, we updated **Table 5** by compute position consistency and fairness separately for **faithfulness**, and **consistency of style** in addition to fluency. We find that the degree of position bias varies across LLMs and criteria. For example, GPT-4o is relatively robust on fluency but less stable on faithfulness and consistency of style, whereas Gemini-2.0-Flash attains stronger position consistency on faithfulness but exhibits noticeably weaker robustness on the other two criteria.
>
> These new results directly address your point and provide a more fine-grained understanding of position bias across the three evaluation dimensions.
>
> ### 3. Prompt adherence and nonsensical answers
>
> > **Question 1:** Was there any cases where the LLMs did not follow the prompt instruction and generated nonsensical answers? If so, how was it handled?
> >
>
> Yes, we occasionally observed nonsensical outputs (`None`) that did not follow our instructions in the experiments. To solve this, we prompt the LLM evaluator again to get a meaningful answer. In the revision, we described this in **Section 4.2**.

---

### Official Review · Reviewer_Kave · 2025-10-29

**Soundness:** 2
**Presentation:** 3
**Contribution:** 1
**Rating:** 2
**Confidence:** 3

**Summary:**

The title promises a fine grained ranking evaluation. While reading the paper it becomes clear that this only means to split the feedback into ‘faithfulness’, ‘fluency’, and ‘consistency of style’. The authors defined style and fluency as a different class, but I would even argue that both are part of the same broader topic which I would have called fluency (maybe naturalness?). This is leaving only two data points per compared translation pair instead of one.
This is much less fine grained than MQM which gives way more detailed information about each sentence and isn’t limited to simply ranking. MQM scores can easily be converted into a ranking, this leaves only the price as an advantage of this method.

**Strengths:**

Releases a ranking benchmark which distinguishes between faithfulness, fluency, and consistency of style.

They analyzed the bias of the LLM evaluators.

The data will be released.

**Weaknesses:**

While having three scores is more fine grained than one, it still provides much less information compared to the MQM granularity.

The authors used language pairs where no existing WMT results are public which would have been easy to compare to.

**Questions:**

Why did you use the comparably small NLLB-200-1.3B model when the results were so weak that you had to downsample it? Using the 3.3B model should not have been any issue from the hardware perspective. If you could run the Qwen2-72B model using the NLLB-MoE-52B should also have been possible.

Why didn’t you use a language pair supported by WMT? That way you could have used submitted results and compared directly to MQM results or other metrics evaluated at WMT. You can find the ende, enes, and jazh data here: https://github.com/google/wmt-mqm-human-evaluation/tree/main/generalMT2024
It’s obviously too late now to change it since you already collected the human rating, but it seems like an odd choice to build a dataset which can not be directly compared to existing data.

Why did you separate ‘consistency of style’ from ‘fluency’, but no subcategories for the ‘faithfulness’ part? I would say that changing the style also breaks the ‘fluency’ (in a broader sense not as defined in the paper). MQM provides a lot more categories, also for ‘faithfulness’ (by MQM called Accuracy).

---

> ### Author Response · Authors · 2025-11-28
> **Response to Reviewer Kave (1/2)**
>
> Thank you for your comments and for pointing us to the WMT MQM data. We address the concerns below.
>
> ### 1. Fine-grained v.s. MQM granularity
>
> > **Summary and Weakness:** This is much less fine grained than MQM which gives way more detailed information about each sentence and isn’t limited to simply ranking. While having three scores is more fine grained than one, it still provides much less information compared to the MQM granularity.
> >
>
> Our use of *“fine-grained”* is **relative to existing LLM-as-a-judge and regression/ranking metrics**, which typically return a single holistic score or ranking. We do not intend to claim that our framework is more fine-grained than full MQM taxonomies.
>
> Our contributions are:
>
> - We move from single holistic ranking evaluation to three widely used criteria.
> - We construct, to our knowledge, the **first benchmark with human-annotated, reference-free *pairwise* rankings** on these criteria (rather than deriving pairwise data by post-hoc conversion from sentence-level error labels).
> - We show that FiRE **outperforms strong automatic MQM-style evaluators** (e.g., error-based metrics) and strong ranking metrics in matching human pairwise decisions, while also enabling criterion-specific diagnostics (e.g., models that are fluent but hallucination-prone).
>
> In addition, MQM is designed for **error-type-level analysis**: human annotators first locate errors and categorize them (e.g., different types of errors), from which one can later synthesize scores. In contrast, FiRE is designed for **criterion-level pairwise decisions** in a reference-free setting. MQM remains richer in error granularity, while FiRE provides scalable, criterion-level pairwise evaluation and a new human-labeled testbed for studying LLM-as-a-judge and bias.
>
> ### 2. Use of NLLB-200-1.3B and downsampling
>
> > **Question 1:** Why did you use the comparably small NLLB-200-1.3B model when the results were so weak that you had to downsample it?
> >
>
> Our goal in constructing the dataset was to **cover a broad spectrum of translation quality**, from relatively weak to very strong systems. For this reason, we intentionally included systems of very different scales and types, from NLLB-200-1.3B through 13B and 72B LLMs, to strong commercial systems.
>
> During pilot annotation, we found that NLLB-200-1.3B produced worse translations than the other systems, making many pairs involving this system extremely easy to judge (annotators consistently regarded it as clearly worse). Such trivial pairs provide limited research value for studying (i) fine-grained preferences and (ii) differences among strong evaluators. Therefore, we downsampled pairs involving NLLB-200-1.3B to prevent these trivial comparisons from dominating the dataset.
>
> This decision was not due to hardware constraints—we could have used larger NLLB variants if needed—but due to **annotation quality and dataset balance considerations**. Our main conclusions are driven by comparisons among modern, strong systems (Qwen2-72B, GPT-4o, DeepL, LanMT, ALMA-13B-R), and are robust to the weaker systems.
>
> In the revision, we updated **Section 3.2** and added a section in **Appendix J** to explain the rationale for including and downsampling NLLB-200-1.3B.

---

> ### Author Response · Authors · 2025-11-28
> **Response to Reviewer Kave (2/2)**
>
> ### 3. WMT language pairs and use of MQM datasets
>
> > Why didn’t you use a language pair supported by WMT? You can find the ende, enes, and jazh data here: https://github.com/google/wmt-mqm-human-evaluation/tree/main/generalMT2024.
> >
>
> In WMT general translation tasks, English→Chinese is an important language pair to evaluate, which has been incorporated in existing results. We agree that connecting to WMT MQM datasets is valuable. Based on your comment (and Reviewer Fo13’s), we have extended our experiments in two directions:
>
> 1. **MQM23 Hebrew→English (HE→EN) and MQM24 Japanese→Chinese (JA→ZH):** We conduct new experiments on the MQM23 HE→EN and MQM24 JA→ZH datasets. Based on the MQM taxonomy, we map fine-grained MQM error tags into our three criteria (faithfulness, fluency, and consistency of style) using the grouping outlined in Table 13, and then apply FiRE on this independent benchmark. This experiment not only provides a bridge between our method and existing MQM datasets, but also offers an additional evaluation in a low-resource language (Hebrew).
> 2. **WMT24++ Japanese→Chinese (JA→ZH):** We additionally evaluate FiRE and other baselines on a JA→ZH test set from WMT24++, using exactly the same annotation protocol described in the paper. On this dataset, we again observe: 1) high inter-annotator agreement across the three criteria. 2) consistent results that FiRE outperforms strong baselines and remains robust across criteria.
>
> In the revision, we updated as follows:
>
> - Add a subsection (**Section 5.4**) that introduces the MQM23 HE→EN and MQM24 JA→ZH experiments and summarizes the main findings.
> - Report the annotated JA→ZH results in the **Appendix L** and **Table 11**, as a complementary language pair to EN→ZH and RU→ZH, and and highlight that our conclusions hold across these four directions (EN→ZH, RU→ZH, HE→EN, JA→ZH).
>
> ### 4. Choice of three criteria
>
> > **Summary and Question 3:** The authors defined style and fluency as a different class, but I would even argue that both are part of the same broader topic which I would have called fluency (maybe naturalness?). Why did you separate ‘consistency of style’ from ‘fluency’, but no subcategories for the ‘faithfulness’ part? I would say that changing the style also breaks the ‘fluency’ (in a broader sense not as defined in the paper). MQM provides a lot more categories, also for ‘faithfulness’ (by MQM called Accuracy).
> >
>
> The classification **faithfulness / fluency / style** follows common practice in machine translation studies, and is also consistent with MQM and related typologies, where Fluency (or Linguistic conventions) and Style are treated as distinct dimensions. Many prior works [1-4] similarly separate these criteria.
>
> Conceptually and empirically, we believe that:
>
> - **Faithfulness** is meant to capture hallucinations and semantic preservation: whether the translation preserves all and only the information in the source (entities, facts, relations).
> - **Fluency** reflects the model’s general proficiency in the target language, including grammar, collocation, and naturalness.
> - **Consistency of style** focuses on **cross-lingual style transfer**, including formality level, tone, and stylistic alignment with the source. A translation can be highly fluent yet stylistically shifted (e.g., informal tone for a formal report), which is why we consider this dimension separately.
>
> In our analyses, these criteria are **not redundant**: different systems show different trade-offs (e.g., some are very fluent but less faithful, others preserve content well but are stylistically less matched), and the style rankings sometimes diverge from fluency rankings. This supports the usefulness of distinguishing them.
>
> In the revision, we added a section in Appendix to justify our choice of criteria.
>
> [1] Qingyu Lu, Baopu Qiu, Liang Ding, Liping Xie and Dacheng Tao. Error Analysis Prompting Enables Human-Like Translation Evaluation in Large Language Models: A Case Study on ChatGPT. ACL 2023.
>
> [2] Zhiwei He, Tian Liang, Wenxiang Jiao, Zhuosheng Zhang, Yujiu Yang, Rui Wang, Zhaopeng Tu, Shuming Shi and Xing Wang. Exploring Human-Like Translation Strategy with Large Language Models. Transactions of the Association for Computational Linguistics 12 (2023): 229-246.
>
> [3] Zhaopeng Feng, Jiayuan Su, Jiamei Zheng, Jiahan Ren, Yan Zhang, Jian Wu, Hongwei Wang and Zuozhu Liu. M-MAD: Multidimensional Multi-Agent Debate for Advanced Machine Translation Evaluation. ACL 2024.
>
> [4] Shuqiao Sun, Yutong Yao, Peiwen Wu, Feijun Jiang and Kaifu Zhang. PMMT: Preference Alignment in Multilingual Machine Translation via LLM Distillation. ArXiv abs/2410.11410 (2024)

---

### Official Review · Reviewer_svgb · 2025-10-31

**Soundness:** 2
**Presentation:** 2
**Contribution:** 3
**Rating:** 4
**Confidence:** 4

**Summary:**

the paper presents a new LLM-based evaluation method, FIRE, by ranking two translations of the same source
three ranking criteria are distinguished: adequacy, fluency and style, as well as overall quality
the rankings are performed by language models on English-Chinese and Russian-Chinese translations and then compared with rankings of human evaluators
the FIRE results are more similar to human judgements than KIWI, xCOMET, metricX and MT-ranker

**Strengths:**

The evaluation on three different criteria provides more insights than only an overall scores

The method is clearly explained

**Weaknesses:**

the data samples apparently consist only of isolated sentences, so that the context is not taken into account
several recent studies have shown that the evaluation should be done on a paragraph level, not on isolated sentences

some parts are not fullly clear (see questions)

**Questions:**

034  why is BLEU (and other similar metrics) called a regression-based metric? What is the regression there?

Maybe the idea is that they are reference-based?
so "similarity-based" or "overlap-based" would be a good description

based on a single score


053: one source sentence: meaning that the evaluation is on isolated sentences, without context


180: which systems are NMT (encoder-decoder) and which are LLMs (decoder only)?

258: why report only DeepSeek-R1 results and not of other used models?

268: aligns with the majority vote of human annotations:
Why not compare each data point with each data point?
Overall result (majority vote) might be similar even though the actual annotations are quite different

395: a figure discussed in the main text should not be in appendix

the organisation of a paper should be in a way that the reader does not need to look into Appendix at all

421: position bias has already been discussed in 4.1

---

> ### Author Response · Authors · 2025-11-28
> **Response to Reviewer svgb (1/2)**
>
> Thank you for your careful reading and constructive comments. We are glad that you find the method clearly explained and appreciate the added insight from evaluating adequacy/faithfulness, fluency, and consistency of style separately. Below we address the weaknesses and questions in detail and indicate the changes we will make in a revised version.
>
> ### 1. Sentence-level v.s. paragraph-level and document-level evaluation
>
> > **Weakness and Question for 053:** Data samples consist of isolated sentences without context; recent work suggests evaluation should be done at paragraph level.
> >
>
> You are right that our current benchmark is segment-based: each data point consists of one source sentence and two translations (**as described in Section 3.2**). This design choice follows the dominant practice in MT meta-evaluation, where most existing ranking benchmarks (e.g., WMT RR08–16, DA, MQM) operate at the segment level, even when document context is available during annotation. Our main goal in this paper is to (i) introduce a fine-grained, criterion-based *paradigm* and (ii) provide the first *reference-free* human-annotated testbed for such pairwise ranking; starting from the standard sentence-level formulation lets us control complexity and systematically compare against existing regression-based, error-based, and ranking-based metrics.
>
> At the same time, we fully agree that paragraph- and document-level evaluation is crucial, especially for discourse phenomena and context-sensitive errors. Importantly, FiRE itself is agnostic to the length of the source: the prompts and decision rule remain unchanged if we feed a paragraph (or document) as the source instead of a single sentence. In the revision, we have updated **Appendix K** and cited some studies to reflect the importance of paragraph-level and document-level evaluation and our future directions.
>
> ### 2. Terminology: “regression-based” vs “overlap-based” metrics
>
> > **Question for 034:** Why is BLEU (and similar metrics) called a regression-based metric? Perhaps “similarity-based” or “overlap-based” is a better description.
> >
>
> Thank you for pointing out this ambiguity. Our intention in Section 2.1 was to distinguish *scalar-score* metrics (which output a single numeric score per translation) from error-based and ranking-based approaches. Within that group, some metrics (e.g., COMET, BLEURT, MetricX) are indeed regression models trained to predict a quality score, while BLEU is a hand-crafted similarity/overlap measure. In the revision, we updated as follows:
>
> - Thank you for your suggestion. We use “**overlap-based**” for hand-crafted metrics such as BLEU and related n-gram or embedding-based similarity measures.
> - Update the corresponding descriptions in **Introduction**, and **Section 2.1** to avoid calling BLEU a regression-based metric.
>
> We hope this resolves the confusion while preserving the conceptual contrast with error-based and ranking-based paradigms.
>
> ### 3. Clarifying which systems are NMT vs LLMs
>
> > **Question for 180:** which systems are NMT (encoder–decoder) and which are LLMs (decoder-only)?
> >
>
> We appreciate this suggestion for clarity. In the revision we added a section and a table in Appendix summarizing, for each system: open/closed-source status and architecture type:
>
> - **Encoder–decoder systems:** NLLB-200-1.3B.
> - **Decoder-only systems:** ALMA-13B-R, Qwen2-72B-Instruct, GPT-4o.
> - DeepL and LanMT are commercial translation systems. Both systems acknowledge that they are transformer-based but the details of architecture are undisclosed.
>
> In the process of data construction, we select these systems to incorporate various hyper-parameter sizes (1.3B, 13B, 72B and larger) and architectures. Thank you for help us further clarify this. In revision, we incorporated the details of MT systems in the **Appendix I** and **Table 9**.
>
> ### 4. Why we mainly report DeepSeek-R1 as the FiRE backbone?
>
> > **Question for 258 and 395:** Why do you report only DeepSeek-R1 results, but not those of the other LLM evaluators? A figure discussed in the main text should not be in Appendix.
> >
>
> Our main tables (e.g., Table 2 and 3) use DeepSeek-R1 as the default FiRE backbone to keep the comparison with regression- and error-based metrics focused and space-efficient. However, we do evaluate FiRE with six additional LLM backbones, and report these results in **Section 5.1, Table 4 and Figure 6 (Appendix)**. In revision, we updated as follows to make this more visible:
>
> - Emphasize earlier in **Section 4.2** that DeepSeek-R1 is chosen as the main backbone for the core results, but that Section 5.1 contains an ablation over seven LLM evaluators (GPT-4o, Claude-3.5-Sonnet, Gemini-2.0-Flash, Qwen2.5-72B-Instruct, Mistral-Large-Instruct, QwQ-32B, DeepSeek-R1).
> - Move Figure 6 from the Appendix into the main body (as Figure 4 for now) to help clarify **Section 5.1**.

---

> ### Author Response · Authors · 2025-11-28
> **Response to Reviewer svgb (2/2)**
>
> ### 5. Majority vote vs per-annotator comparison
>
> > **Question for 268:** Why not compare each data point with each data point? Overall result (majority vote) might be similar even though the actual annotations are quite different.
> >
>
> We follow the common practice in MT studies of using the **majority vote** as the gold label for meta-evaluation. Individual annotations are noisy, and using the majority vote reduces variance and provides a more stable target, especially in our 3-annotator, 3-class setting. This is also consistent with how we compute Fleiss’ kappa in **Table 1**, where the aggregated labels summarize substantial inter-annotator agreement.
>
> We appreciate your insightful suggestion. We have additionally computed agreement between each evaluator and **each individual annotator**, and observed that the results are consistent with the majority vote.
>
> In the revision, we added a short paragraph as **Appendix M** and **Table 12** reporting these per-annotator agreements, and explicitly state that our conclusions are robust to this choice of gold labels.
>
> ### 6. Redundancy in the description of position bias
>
> > **Question for 421:** Position bias has already been discussed in Section 4.1.
> >
>
> Thank you for pointing out the redundancy. We first introduce position consistency and fairness as metrics in **Section 4.3**, and then repeat their definitions when analyzing bias in **Section 5.3**. In the revision, we updated as follows to make structure cleaner:
>
> - Keep the **formal definitions** of position consistency and fairness in **Section 4.3 (Metrics).**
> - In **Section 5.3**, refer back to these definitions and focus only on the interpretation of the results (e.g., which evaluators are more or less position-biased), avoiding repeated descriptions.

---

### Official Review · Reviewer_GbcT · 2025-11-02

**Soundness:** 2
**Presentation:** 2
**Contribution:** 3
**Rating:** 4
**Confidence:** 4

**Summary:**

This paper proposes FiRE, a machine translation evaluation framework which focuses on reference free pairwise ranking across broad categories (faithfulness, fluency and style consistency). A benchmark is created and various evaluation methods are analyzed against that benchmark.

**Strengths:**

* The proposed framework is clear and conceptually intuitive and simple
* The benchmark can be a good contribution for subsequent evaluations of MT systems
* Experimentation and ablations appear reasonably thorough

**Weaknesses:**

I have several doubts/questions about the methodology. I'm listing here the main questions but see also the questions below.
* on the benchmark creation, is there a mechanism for quality control? I didn't find enough details on the human annotators and any guardrails for quality.
* I believe Section 4.5 may be one of the main claims of the paper, i.e. that the proposed framework is better than prior ones. However, that section is very small and lacks details for each row presented in Table 3.

Given the amount of uncertainties and questions, I'm currently leaning towards a weak reject and would encourage the authors to provide more details.

**Questions:**

204-205: “our 3-annotator 3-class setting (which typically yields lower κ values) shows comparably substantial inter-annotator reliability (κ = 0.57 − 0.81).”: do you have an intuition or explantation about this?

Table 1: why are there different number of pairs across categories and why not only retain pairs with all 4 categories?

244-246: “To derive the overall pairwise judgment, we further aggregate all errors across the three criteria into a single composite error-based score.”: what is the aggregation method?

201: “three annotators evaluate the pairwise comparisons”: I believe it is important to provide more details on the annotators, the guidelines given to them and quality control process for the data collection. Otherwise, it puts into question the validity of the benchmark created.

Was position consistency also checked for human evaluators? If not, it could be valuable information.

------------- Typos/Presentation

Figure 1: using gloss for the Chinese text would be good to better understand the explanation.

116:  “semi-automate this process using PLMs”: introduce acronym

184-185: “ and three closed-source systems (GPT4o1 , DeepL, LanMT)”: indicate the dates for all closed source systems, not only gpt-4o

306: “The aggregation procedure is described in Appendix D.”: In my opinion the procedure should be described in the main body of the paper, not the appendix as this is an important detail.

---

> ### Author Response · Authors · 2025-11-28
> **Response to Reviewer GbcT (1/2)**
>
> Thank you for your careful reading and constructive comments. We appreciate your accurate summary of our work, your recognition of the clarity and simplicity of the framework. and your acknowledgement of the value of our contribution. Below we address the weaknesses and questions in detail and indicate the changes we will make in a revised version.
>
> ### 1. Annotation quality control
>
> > **Weakness 1 and Question for 204-205 and Question for 201:** mechanism for quality control? I didn't find enough details on the human annotators and any guardrails for quality. “our 3-annotator 3-class setting (which typically yields lower κ values) shows comparably substantial inter-annotator reliability (κ = 0.57 − 0.81).”: do you have an intuition or explantation about this?
> >
>
> Thank you for your careful review and asking about the quality control and annotation process of our benchmark.
>
> - **Annotator qualifications:** The background and language proficiency of our annotators are described in Section 3.2 (around line 206) as well as in **Appendix E** and **Table 7**, where we report their CEFR levels and relevant details.
> - **Annotation guidelines:** The annotation guidelines consist of two main components. The first provides an overview of the annotation task, including the input format and the overall objective. The second specifies the operational definitions of the three evaluation criteria, accompanied by illustrative examples. In the revision, we added the details of annotation guidelines in **Appendix C**.
> - **Quality control procedures:** Before formal annotation, we conducted several pilot annotation rounds to calibrate annotators’ understanding and ensure consistency. We also believe that the pairwise ranking setup, especially with explicit fine-grained criteria (faithfulness, fluency, and consistency of style), simplifies the task and reduces ambiguity compared to absolute scoring, which contributes to higher inter-annotator agreement. We have updated the **Section 3.2** to reflect this quality control procedure.
>
> Regarding the relatively strong κ values (0.57–0.81) in our 3-annotator 3-class setting, our intuition is that pairwise comparison is cognitively simpler and more stable than multi-level scoring, and that breaking the evaluation into explicit criteria further reduces subjectivity and variance across annotators. In the revision, we added this explanation in **Section 3.2**.
>
> ### 2. Logic of this paper
>
> > **Weakness 2:** I believe Section 4.5 may be one of the main claims of the paper, i.e. that the proposed framework is better than prior ones. However, that section is very small and lacks details for each row presented in Table 3.
> >
>
> Thank you for your helpful comments on the organization of our experimental sections. In the revision, we have updated **Section 4.5** with concrete comparison. In addition, we would like to mention that  Section 4.5 is not intended to stand alone, but to function as part of a coherent evaluation pipeline. Since our work proposes a general evaluation framework, we compare against a broad range of prior paradigms (regression-based, error-based, and ranking-based), rather than restricting the scope only to fine-grained evaluation, which is still an emerging task. Sections 4.4–4.6 are therefore designed as a smooth progression, together forming the core experimental evidence of our paper. We have updated **Section4.4-4.6** to strengthen the logic and make the connection more coherent.
>
> ### 3. Data part
>
> > **Question for Table 1:** why are there different number of pairs across categories and why not only retain pairs with all 4 categories?
> >
>
> To preserve data diversity and coverage, we did not restrict the dataset to only the intersection of samples across all criteria. Specifically, for benchmark construction, we first collected annotations for all 1,600 pairwise items under each criterion and computed the inter-annotator agreement (Fleiss’ κ) on the full set to assess annotator alignment. We then removed cases where all three annotators selected different labels, since such instances do not yield a reliable consensus and therefore cannot be used for evaluation. As a result, the number of retained pairs may vary slightly across different categories in Table 1. In revision, we supplement this information in **Section 3.2**.

---

> ### Author Response · Authors · 2025-11-28
> **Response to Reviewer GbcT (2/2)**
>
> ### 4. Aggregation Method
>
> > **Question for 244-246:** “To derive the overall pairwise judgment, we further aggregate all errors across the three criteria into a single composite error-based score.”: what is the aggregation method?
> >
>
> Thank you for pointing this out and apologize for the lack of clarity in our description. For the error-based evaluators (M-MAD and GEMBA-MQM), we first map their fine-grained error types to three criteria (Faithfulness, Fluency, and Consistency of Style) following the procedure described in Table 13. This yields three criterion-specific MQM-style scores for each translation. We then aggregate these three scores by simple summation to obtain a single composite error-based score, which is subsequently used to derive the overall pairwise judgment between the two candidates. We have revised the relevant description in **Section 4.1** the updated version to make this aggregation process explicit and clearer.
>
> ### 5. Human Position Consistency
>
> > **Question 5:** Was position consistency also checked for human evaluators? If not, it could be valuable information.
> >
>
> Thank you for your this interesting question. We did not conduct a dedicated position consistency analysis for human annotators. Our analysis focuses on LLM evaluators because, unlike humans, they do not have long-term memory of previously seen samples, making position bias a more intrinsic modeling issue. In contrast, human annotators may implicitly remember sentences or earlier judgments during the annotation process, which could introduce confounding effects (e.g., recall or learning bias) when re-presenting the same pairs in reversed order. To avoid such potential contamination and ensure fair experimental conditions, we therefore did not conduct a detailed position consistency study on human annotators in this work. In the revision, we have updated **Appendix K** to include the discussion about the position bias of human annotators.
>
> ### 6. Typo
>
> Thank you for pointing out these typos. In the revision, we updated as follows:
>
> - Add explanation in **Figure 1** to help reader understand the example.
> - Revise the sentence to introduce acronym for PLMs.
> - Incorporate details of MT systems in **Appendix I** and cited in **Section 3.2**.
> - Move aggregation procedure into main body (**Section 4.4**).

---

### Meta-Review · Area_Chair_b9Yg · 2026-01-07

**Summary:**

This paper proposes FiRE, a fine-grained pairwise ranking framework for machine translation evaluation, assessing the quality by faithfulness, fluency, and consistency. The authors introduce a new human-annotated benchmark and conduct extensive evaluations comparing FiRE against existing automatic metrics and LLM-based evaluators. Reviewers generally agree that the framework is well-written, the method is sound. The released benchmark could be a useful resource for future MT evaluation.

The major concerns are benchmark quality, soundness of the evaluation method by three dimensions, and insufficient experiments.

**Reviewer Concerns:**

- Benchmark validity and method clarity: Multiple reviewers request more detail on human annotation procedures, annotator background, guidelines, aggregation methods, and quality control. Important aspects of the methodology (e.g., aggregation of criteria, benchmark construction details). Though the authors explained in the rebuttal, inter-annotator reliability (κ = 0.57 − 0.81) is still a concern given that the benchmark is one of the main contributions of this work.

- Reviewer Kave strongly questions whether splitting evaluation into three criteria truly constitutes fine-grained evaluation, arguing that MQM provides far richer diagnostic information and that FiRE’s advantage is mainly cost. The paper would benefit from clearer positioning relative to MQM and a more explicit justification of design choices (e.g., why style is separated from fluency, but faithfulness is not further decomposed).

- The evaluation is limited to sentence-level, high-resource language pairs, newly collected data, and small NLLB-200-1.3B model. Reviewers note to validate FiRE on existing MQM/WMT datasets or paragraph-level contexts, which would strengthen validity.

**Reviewer Scores:**

4.5

---

### Decision · Program_Chairs · 2026-01-26

Reject